# Identification of a family of species-selective complex I inhibitors as potential anthelmintics

Taylor Davie[1,2], Xènia Serrat[1,2], Lea Imhof[3,4], Jamie Snider[1,2], Igor Štagljar [1,2,5,6], Jennifer Keiser[3,4], Hiroyuki Hirano [7], Nobumoto Watanabe [7], Hiroyuki Osada [7,8] & Andrew G. Fraser [1,2] ✉

Soil-transmitted helminths (STHs) are major pathogens infecting over a billion people. There are few classes of anthelmintics and there is an urgent need for new drugs. Many STHs use an unusual form of anaerobic metabolism to survive the hypoxic conditions of the host gut. This requires rhodoquinone (RQ), a quinone electron carrier. RQ is not made or used by vertebrate hosts making it an excellent therapeutic target. Here we screen 480 structural families of natural products to find compounds that kill *Caenorhabditis elegans* specifically when they require RQ-dependent metabolism. We identify several classes of compounds including a family of species-selective inhibitors of mitochondrial respiratory complex I. These identified complex I inhibitors have a benzimidazole core and we determine key structural requirements for activity by screening 1,280 related compounds. Finally, we show several of these compounds kill adult STHs. We suggest these species-selective complex I inhibitors are potential anthelmintics.

Soil-transmitted helminths (STHs) are major pathogens of humans and livestock[1,2]. Over a billion humans are infected by STHs including roundworm (*Ascaris lumbricoides*), hookworm (*Necator americanus* and *Ancylostoma duodenale*), and whipworm (*Trichuris trichiura*) and helminth infections are responsible for multiple neglected tropical diseases (NTDs). These infections result in malnutrition, malaise and weakness, and can cause developmental defects and impaired growth in children[3]. In addition, STHs infect a high proportion of livestock leading to reduced yield. This is a particular problem in poorer communities where such losses can have major health and economic consequences. While there are excellent frontline anthelmintics including benzimidazoles (e.g., albendazole and mebendazole)[4,5] and macrocyclic lactones (e.g., ivermectin)[6,7], there are few classes of commercial anthelmintics and resistance to these drugs is widespread

in livestock and cases of reduced efficacy have been observed in human parasites as well[8,9]. There is thus an urgent need for new classes of anthelmintic drugs to control and treat these major pathogens which present a key challenge in global health.

Any effective anthelmintic must target the STH without harming the vertebrate host. One way to do this is to develop drugs that target a process that is essential for the STHs but absent from the host and that is the approach we take here. We focus on a unique aspect of STH metabolism—rhodoquinone-dependent metabolism[10–13]. During the stages of the lifecycle when the STH lives in the soil outside the host, it generates energy using oxidative phosphorylation. Electrons enter the mitochondrial electron transport chain (ETC) either at complex I or via a number of quinone-coupled dehydrogenases that include complex II, succinate dehydrogenase. They are first transferred to ubiquinone

[1]The Donnelly Centre, University of Toronto, 160 College Street, Toronto M5S 3E1, Canada. [2]Department of Molecular Genetics, University of Toronto, Toronto, Ontario, Canada. [3]Swiss Tropical and Public Health Institute, Kreuzstrasse 2, CH-4123 Allschwil, Switzerland. [4]University of Basel, CH-4000 Basel, Switzerland. [5]Mediterranean Institute for Life Sciences, Meštrovićevo Šetalište 45, HR-21000 Split, Croatia. [6]Department of Biochemistry, University of Toronto, Toronto, Ontario, Canada. [7]Chemical Resource Development Research Unit, RIKEN Center for Sustainable Resource Science, 2-1 Hirosawa, Wako Saitama 351-0198, Japan. [8]Institute of Microbial Chemistry (BIKAKEN), 3-14-23 Kamiosaki, Shinagawa-ku, Tokyo 141-0021, Japan. ✉e-mail: andy.fraser@utoronto.ca

(UQ) and then pass through complex III and complex IV where they ultimately are accepted by molecular oxygen as the terminal electron acceptor. As the electrons flow through the ETC, protons are translocated across the mitochondrial membrane, establishing a proton motive force which powers ATP synthesis by the F0F1-ATP synthase, complex V (Fig. 1a). This UQ-using oxidative ETC is identical between host and parasite in aerobic conditions. However, many STHs must survive extended periods in the highly anaerobic environment of the host gut—adult *Ascaris*, for example lives for months in these conditions. Without available oxygen as the terminal electron acceptor, the STHs cannot use the aerobic UQ-coupled ETC. However, STHs have a key adaptation that still allows them to use a rewired form of the ETC and it is this that provides the potential target for anthelmintics. Electrons still enter the ETC from NADH into complex I and onto the quinone pool but instead of flowing through to complex IV, they exit the ETC at complex II which now acts as a fumarate reductase rather than a succinate dehydrogenase (Fig. 1b). This allows fumarate to be used as a terminal electron acceptor. Crucially, UQ cannot be used as an electron carrier to power the fumarate reductase activity—instead STHs use rhodoquinone (RQ), a highly related quinone[14–16]. This basic pathway (often also termed 'malate dismutation') was first described over 50 years ago in a series of elegant biochemical studies (reviewed in refs. 10,11,17). More recently, the pathway for RQ synthesis and the key molecular switch that determines whether UQ or RQ is made were identified in the genetically tractable nematode *C. elegans*[18–20] and thus we now know the way the electron transport chain is rewired, how the key electron carrier RQ is made, and many of the genes required.

From the perspective of anthelmintic discovery, RQ synthesis and RQ-dependent metabolism is a very attractive target. RQ is only made and used by a small number of animal species: nematodes, platyhelminths, annelids, and molluscs. Since STHs make and use RQ but vertebrate hosts do not, RQ synthesis and RQ-dependent metabolism provide a critical target that differs between host and parasite. If we could identify drugs that specifically target RQ synthesis or RQ-dependent metabolism, they should kill the parasite without affecting the host—this is our goal.

The rewired ETC that is used for RQ-dependent metabolism has three clear attack points for drugs: complex I, the sole entry point for electrons and the sole source of proton pumping; complex II, the critical exit point for electrons; and the pathway for RQ synthesis (schematic in Fig. 1c). We previously showed that the free-living nematode *C. elegans* makes and uses RQ[18] and that inhibitors of complexes I or II, or mutation of the RQ synthesis pathway (the three main attack points) all block RQ-dependent metabolism in *C. elegans*. *C. elegans* is thus an excellent model for RQ-dependent metabolism and it should be possible to carry out high throughput drug screens in vivo in *C. elegans* to identify additional inhibitors of RQ-dependent metabolism.

In this study, we screened a collection of natural products comprising 480 distinct structural classes of compounds as well as their derivatives[21,22] for compounds that affect our *C. elegans* model of RQ-dependent metabolism—this covers the broad chemical space of 25,000 individual natural products. We identified multiple structural groups of potential RQ-dependent metabolism inhibitors and showed that several of these have highly specific effects on individual ETC complexes. These include a family of compounds with a benzimidazole core that we show can specifically and potently target complex I—this is an unexplored activity for benzimidazole anthelmintics. We screened 1,280 benzimidazoles and identified a small structurally-related subset that shows good species selectivity, inhibiting *C. elegans* complex I with >10-fold lower concentrations than bovine or murine complex I. Several of these also have no detectable effect on growth in normoxia. Finally, we show that several of these benzimidazole complex I inhibitors are active against *Heligmosomoides polygyrus*, a mouse STH—their effect is most potent in adult stages which contain mainly RQ and rely on RQ for survival in the host gut. We have thus identified a distinct family of species-selective complex I inhibitors that potently kill *C. elegans* and *H. polygyrus* under conditions where they require RQ-dependent metabolism for survival. These are potential anthelmintics that may similarly kill STHs in the host gut where they require RQ-dependent metabolism.

## Results

### A screen in *C. elegans* for natural products that inhibit RQ-dependent metabolism

*C. elegans* is highly related to STHs[23] and *C. elegans* makes and uses RQ[18,20]. It is thus an excellent system for screens to identify compounds that specifically block RQ synthesis and RQ-dependent metabolism. We previously showed that *C. elegans* requires RQ synthesis and RQ-dependent metabolism to survive exposure to potassium cyanide (KCN) for 15 h[18]. This observation is the basis for a previously published image-based movement assay for RQ-dependent metabolism. In outline, we treat *C. elegans* L1 larvae with 200 μM KCN for 15 h—wild-type worms that can produce RQ are immobile but alive and if we remove the KCN by dilution they rapidly recover movement (Fig. 1d). However, worms that cannot make RQ (e.g., that have mutations in the kynurenine pathway (Fig. 1d)) or that cannot carry out RQ-dependent metabolism (e.g., due to inhibition of complex I or complex II) are dead after 15-h KCN treatment and hence no recovery of movement is seen (Fig. 1e). This was first shown in Del Borrello et al.[18] and we confirm those results here. This allows a simple drug screen for products that specifically kill *C. elegans* when they require RQ-dependent metabolism for survival. Worms are treated with 200 μM KCN with either test compounds or DMSO controls and KCN is removed after 15 h, and movement measured 3 h later. If worms recover movement similar to controls, then the compound does not affect RQ-dependent metabolism. However, if worms are dead after 15-h KCN treatment in the presence of the test compound, no recovery is seen—that compound is thus a potential inhibitor of RQ-dependent metabolism. Finally, to ensure that the compound's effect is specific for RQ-dependent metabolism and not some more general effect on worm viability, we also examine the effect of each compound on the growth and viability of worms in normoxia where they do not require RQ and do not use RQ-dependent metabolism. If a compound has little effect on growth and viability in normoxia but is strongly lethal in our RQ-dependent metabolism assay, this is a potential specific inhibitor of RQ synthesis or RQ-dependent metabolism. Our goal here is to identify such compounds and characterize their effects on RQ-dependent metabolism—they are anthelmintic candidates that act via inhibition of the anaerobic RQ-dependent metabolism that allows STHs to survive in the host gut.

We screened a library of 480 natural products and their derivatives from the RIKEN Natural Product Depository (RIKEN NPDepo)[21,22] in our KCN-based assay for RQ-dependent metabolism and also in a more traditional *C. elegans* normoxic growth and viability assay. 80 of these natural products (Authentic library) were known molecules with previously characterized biological activities. The remaining 400 compounds (Pilot library) are a subset of a larger library of ~25,000 diverse natural product derivatives (NPDs)—each compound screened is thus a representative of a family of structurally-related NPDs. This allows a rapid initial primary screen covering the broad structural space of the entire natural product collection and, once hits are identified, structurally-related NPDs are then screened in secondary assays (schematic in Fig. 2a). This helps define the structural requirements for the biological effects of the primary hits as well as identifying any related compounds with greater potency. The overall results are shown in Fig. 2b. For both assays, each compound was screened in triplicate at 50 μM and the mean value was expressed as a modified z-score. Compounds with modified z-scores < −3 (i.e., >3X median absolution deviation (MAD)) were identified as hits; all data are shown in Table 1. We conduct this primary screen at a high test compound

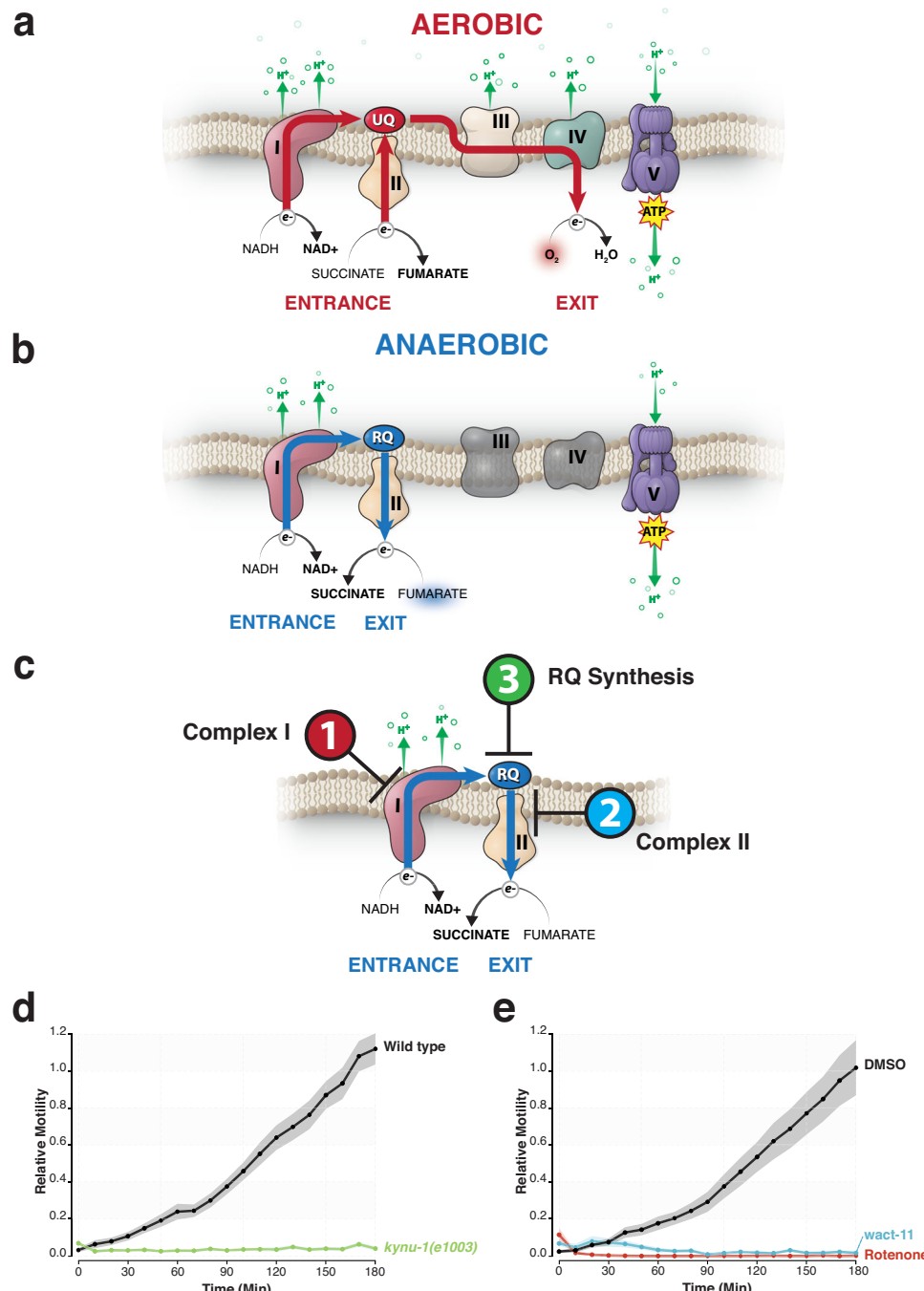

**Fig. 1 | The distinct rhodoquinone-dependent metabolism of parasitic helminths is an attractive target for anthelmintic development. a** Ubiquinone-coupled aerobic electron transport chain (ETC). Electrons generated from the oxidation of NADH and succinate enter the ETC through complexes I and II. Electrons are first transferred to the electron carrier, ubiquinone (UQ) and are ultimately shuttled through complexes III and IV, where they exit the ETC through the reduction of $O_2$. Electron-coupled proton translocation at complexes I, III and IV establishes the proton motive force necessary to drive ATP synthase. **b** Rhodoquinone-coupled anaerobic ETC (Rhodoquinone-dependent metabolism). Electrons from the oxidation of NADH still enter the ETC through the activity of complex I, but are transferred to an alternative electron carrier, rhodoquinone (RQ). In contrast to aerobic conditions, RQ shuttles electrons to complex II, where acting in reverse as a fumarate reductase, fumarate is used as the terminal electron acceptor instead of oxygen. Complex I acts as the sole proton pump to establish the proton gradient to drive the synthesis of ATP by ATP synthase. **c** Key targets for inhibitors of RQ-dependent metabolism. Compounds capable of targeting complex I (sole electron entry point, sole proton pump), complex II (key electron exit point), and RQ synthesis (anaerobic specific electron carrier) are likely to act as anthelmintic candidates. **d** L1 wild-type and *kynu-1* (RQ-deficient) mutant worms treated with 200 μM KCN for 15 h. Survival and overall health of worms were assessed relative to untreated controls by monitoring worm motility over the course of 3 h following the removal of KCN. **e** L1 wild-type worms treated with 12.5 μM rotenone (complex I inhibitor) or 25 μM wact-11 (complex II inhibitor) in combination with 200 μM KCN for 15 h. Survival and overall health of worms were assessed relative to DMSO controls by monitoring worm motility over the course of 3 h following the removal of KCN. All data are the mean of at least three biological replicates; shaded regions are SEM.

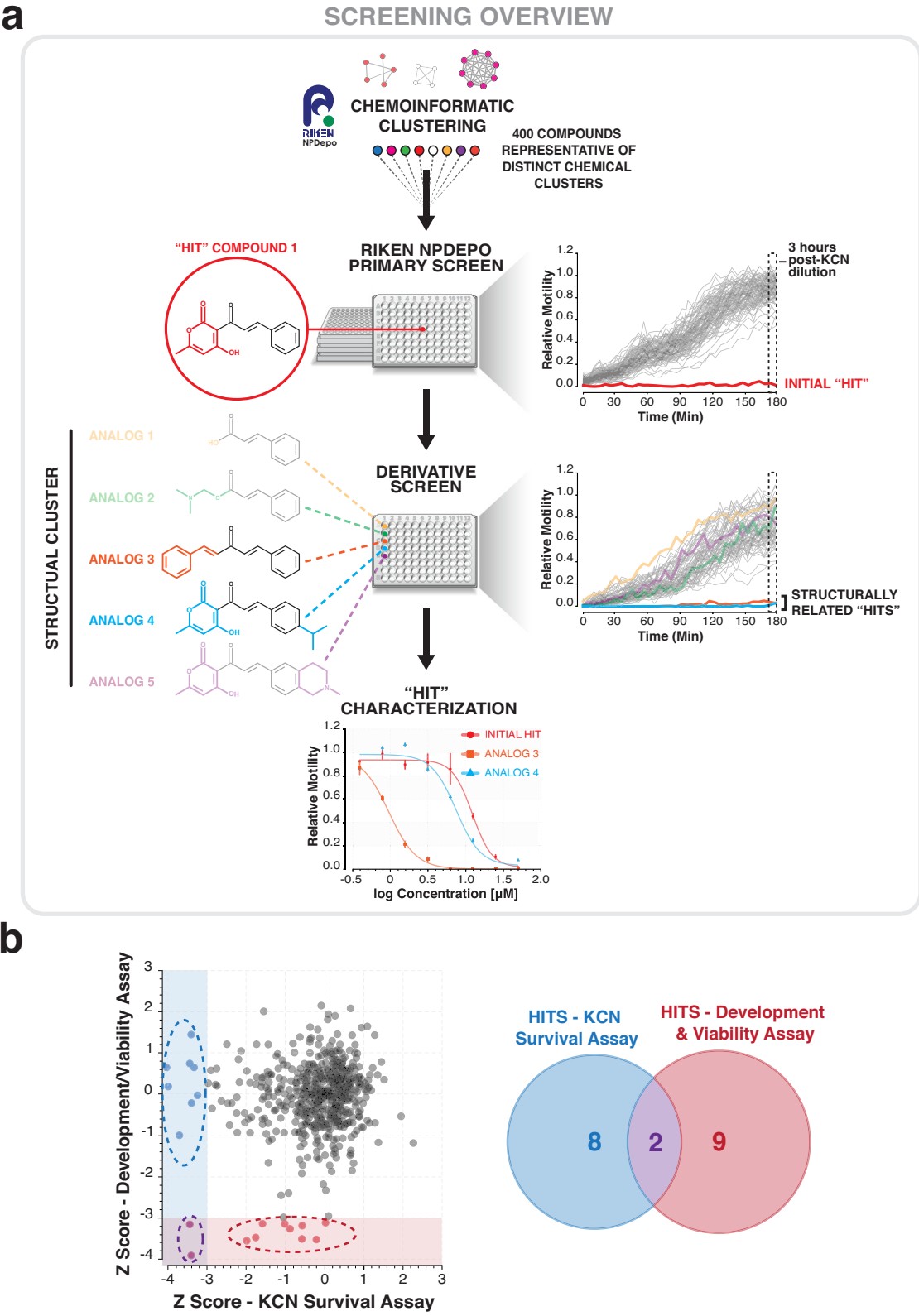

**a** SCREENING OVERVIEW

**b**

concentration of 50 μM to reduce false negatives—we note that *C. elegans* drug screens typically use high primary screen drug concentrations as the worm has powerful xenobiotic responses that often reduce drug accumulation[24].

We identified 9 compounds that only affect normoxic growth. Five of these were natural products with known activities and included nocodazole, an inhibitor of microtubule polymerization[25];

staurosporine, a broad-spectrum kinase inhibitor[26]; camptothecin, a DNA topoisomerase inhibitor[27]; reveromycin A, an isoleucyl-tRNA synthetase inhibitor[28]; and aristolochic acid, a known mutagen[29]. These drugs all target core essential cellular components and the observed lethality is consistent with this. We also identified four uncharacterized compounds that inhibited *C. elegans* growth and viability. Several show no activity in HEK293 cells and these are potential anthelmintics.

**Fig. 2 | Screens to identify inhibitors of RQ-dependent metabolism. a** Overview of the screening strategy for RIKEN Pilot library. The Pilot library consists of 400 compounds, each representative of a cluster of related molecules and analogs from a diverse collection of ~25,000 natural products and derivatives. The initial 400 molecules were screened in two assays of *C. elegans* viability. Analogs, derivatives, and related molecules within the same structural cluster of each hit molecule were provided for additional screening–expanding individual hits into structural groups. Structural groups of molecules were tested in secondary assays to identify group members with the greatest bioactivity and/or selectivity. **b** RIKEN libraries of natural products (Authentic and Pilot) were screened at 50 µM in triplicate in two assays of *C. elegans* viability: KCN survival assay and liquid development/viability (see Methods). Each dot represents a single NPD and is the mean of three biological replicates in each respective assay. Data are presented as modified z-scores, calculated using the median and median absolute deviation (MAD) of each chemical plate. 8 NPDs were identified as hits in the KCN survival assay alone; 9 NPDs were identified as hits in the development and viability assay alone; and 2 NPDs were identified as hits in both assays (z-score < −3).

However, for the purposes of this study, we focused on the 8 compounds that showed little effect on growth in normoxia (z-scores > −3) but killed worms specifically when they were relying on RQ-dependent metabolism for survival. These are candidate RQ-dependent metabolism inhibitors and we refer to these as 'RQ-dependent metabolism hits'.

As described above, the full natural product library contains two groups of compounds: the larger group (Pilot library) is of NPDs with no previous reported bioactivity, and a second smaller group (Authentic library) includes compounds that had previously been shown to be active in at least one previous screen. This latter group yielded three potent RQ-dependent metabolism hits, two of which are known inhibitors of oxidative phosphorylation. Niclosamide is a salicylanilide anthelmintic used to treat tapeworm infections[30]. Niclosamide acts as an ionophore and exerts its lethal effects by collapsing mitochondrial membrane potential[31]. We had previously found that other salicylanilide anthelmintics such as closantel are powerful inhibitors of RQ-dependent metabolism (Supplementary Fig. 1) and thus this hit is expected. The second hit, siccanin, has been reported to target complex II in fungi, bacteria, and mammals[32], and complex II is one of the key attack points of RQ-dependent metabolism (see Fig. 1c schematic). Consistent with this previously reported activity, we found siccanin also potently inhibits *C. elegans* complex II in vitro and shows strong activity within our RQ-dependent metabolism assay similar to other known complex II inhibitors (Supplementary Fig. 2). Thus 2 RQ-dependent metabolism hits in the characterized NPD set, niclosamide and siccanin, had clear roles in electron transport and mitochondrial function validating the relevance of our RQ-dependent metabolism hits. We turned our attention to the previously uncharacterized natural products.

## NPD screens identify a distinct class of benzimidazole complex I inhibitors

We identified 7 NPDs as RQ-dependent metabolism hits that had no previously described bioactivity. We screened (also at 50 µM) an additional 137 compounds from the RIKEN NPDepo that represented analogs and/or derivatives of these 7 RQ-dependent metabolism hits and identified 15 structurally related compounds that also inhibited *C. elegans* survival in KCN (Supplementary Fig. 3 and 4). Since complex I and complex II are key attack points for RQ-dependent metabolism, we first tested whether these NPDs directly affect the electron transport chain in vitro using a range of enzymatic assays in purified *C. elegans* mitochondria (see Supplementary Fig. 5). Several of the RQ-dependent metabolism hits affect more than one complex (e.g., anacardic acid is a potent inhibitor of complexes III and IV) and others have no consistent effect on any specific complex (see Table 2 for data). However, one structural group of RQ-dependent metabolism hits (Fig. 3a) showed highly specific inhibition of complex I (Table 2). These compounds are structurally related to papaverine, an alkaloid antispasmodic drug which has also been shown to be able to inhibit complex I[33,34] and this group of compounds act as potent complex I inhibitors in our hands (Fig. 3b). Complex I is the sole entry point for electrons into the ETC during RQ-dependent metabolism and we previously showed that rotenone, a complex I inhibitor, can block RQ-dependent metabolism (Fig. 1e). Three of this cluster of NPDs are highly related to papaverine

but the fourth NPD in the cluster, NPD8790, is structurally distinct and has a benzimidazole core instead of an isoquinoline core. We found that papaverine, and the other isoquinoline-based NPDs, have no significant species selectivity for complex I (Fig. 3c and Supplementary Table 1). NPD8790, however, shows marked species selectivity, inhibiting *C. elegans* complex I ($IC_{50}$ = 2.4 µM) more potently than complex I from either bovine ($IC_{50}$ = 27.2 µM) or murine heart mitochondria ($IC_{50}$ = 28.8 µM)(Fig. 3d and Supplementary Table 1). In addition, we tested NPD8790 on mitochondria purified from several other nematode species (*Caenorhabditis briggsae*, *Pristionchus pacificus* and *Phasmarhabditis hermaphrodita*) as well as mitochondria purified from HEK293 human embryonic kidney cells and find the $IC_{50}$ of NPD8790 against complex I is similar for all nematode species (*C. briggsae*, $IC_{50}$ = 1.6 µM; *P. pacificus*, $IC_{50}$ = 2.1 µM; *P. hermaphrodita*, $IC_{50}$ = 1.1 µM) but human mitochondria have a greater $IC_{50}$ value ($IC_{50}$ > 75 µM), similar to that seen in bovine and murine mitochondria (Supplementary Fig. 6). This is promising for a potential anthelmintic: complex I is used by both host and STH and thus a molecule that selectively targets STH complex I would be able to kill STHs with minimal adverse effects on the host.

NPD8790 has a benzimidazole core and benzimidazoles are well characterized front line anthelmintics that include mebendazole and albendazole[4,5]. The mode of action of known benzimidazole anthelmintics is well established: they target the beta-tubulin BEN-1[4,35–37] and prevent microtubule polymerization. However, we found NPD8790 has a very different activity: inhibition of complex I. We therefore wanted to examine whether known benzimidazoles also affect complex I and whether NPD8790, the benzimidazole that we identified as a complex I inhibitor, also targets BEN-1. We first tested whether any of the commercial benzimidazoles (e.g., fenbendazole in Fig. 4a, b and others in Supplementary Fig. 7) have similar effects to NPD8790 in our RQ-dependent metabolism assay or in normoxia. We find that while NPD8790 kills worms potently when they require RQ-dependent metabolism for survival, commercial benzimidazoles have no effect in the same RQ-dependent metabolism assay. Conversely, commercially available benzimidazole anthelmintics potently kill *C. elegans* in normoxia (Fig. 4b and Supplementary Fig. 7) whereas NPD8790 has very little effect on *C. elegans* normoxic growth until very high concentrations (greater than 500 µM). These data suggest that commercial benzimidazoles and NPD8790 have a different mode of action. Consistent with this, we find that commercial benzimidazoles have no detectable effect on complex I activity when assayed on purified *C. elegans* mitochondria (Fig. 4c). These data suggest NPD8790 targets complex I but commercial benzimidazoles do not. We note that while some early studies suggested a possible role for benzimidazole anthelmintics in affecting the electron transport chain[38–40], these compounds were all subsequently shown to act by inhibition of microtubule polymerization and in our hands none of these has any effect on complex I activity. We also note that different groups reported different electron transport chain targets and, in addition, some of these early studies also found that levamisole (a known nicotinic acetylcholine receptor agonist) affected the electron transport chain[39], suggesting that these studies had a significant rate of false positives in their assays. We thus would use caution interpreting some of these early findings and suggest that our findings here are the first

**Table 1 | C. elegans chemical screening results—summary of NPD bioactivity**

| Compound ID | Library | Hit Type | Structural Group | Bioactivity | | |
| --- | --- | --- | --- | --- | --- | --- |
| | | | | C. elegans KCN Survival Assay $LC_{50}$ (µM) | C. elegans Dev. & Viability Assay $LC_{50}$ (µM) | HEK293 Cell Viability Assay $LC_{50}$ (µM) |
| Rotenone | Control | NA | – | 0.86 | 3.39 | <0.39 |
| Wact-11 | Control | NA | – | 4.43 | 0.42 | >50 |
| Siccanin | Authentic | KCN | – | 6.34 | 15.23 | 22.21 |
| Niclosamide | Authentic | KCN | – | 0.095 | >1000 | – |
| Flunarizine | Authentic | KCN | – | 21.98 | >200 | 30.23 |
| Nocodazole | Authentic | Dev. & Viability | – | – | – | – |
| Staurosporine | Authentic | Dev. & Viability | – | – | – | – |
| Camptothecin | Authentic | Dev. & Viability | – | – | – | – |
| Reveromycin A | Authentic | Dev. & Viability | – | – | – | – |
| Aristolochic acid | Authentic | Dev. & Viability | – | – | – | – |
| NPD6621 | Pilot | KCN | 1 | 48.03 | – | – |
| NPD8902 | Deriv./Analog | KCN | 1 | 28.26 | >100 | 23.54 |
| NPD6380 | Pilot | KCN | 2 | 14.35 | >100 | 29.20 |
| NPD8034 | Deriv./Analog | KCN | 2 | 11.97 | >100 | 11.15 |
| NPD6383 | Deriv./Analog | KCN | 2 | 22.24 | – | – |
| NPD601 | Deriv./Analog | KCN | 2 | 13.60 | – | – |
| NPD1450 | Deriv./Analog | KCN | 2 | 35.28 | >100 | >50 |
| NPD8366 | Pilot | KCN | 3 | 11.62 | >100 | 19.73 |
| NPD8582 | Deriv./Analog | KCN | 3 | 14.70 | – | – |
| NPD6240 | Deriv./Analog | KCN | 3 | 7.60 | >100 | 24.76 |
| FSL0005 | Deriv./Analog | KCN | 3 | 0.93 | 4.18 | 7.48 |
| NPD8298 | Deriv./Analog | KCN | 3 | 2.57 | >100 | >50 |
| NPD10504 | Pilot | KCN | 4 | 15.59 | >100 | 36.63 |
| NPD3577 | Deriv./Analog | KCN | 4 | 15.95 | >100 | 21.43 |
| NPD6303 | Deriv./Analog | KCN | 4 | 32.20 | >100 | 33.35 |
| NPD8790 | Deriv./Analog | KCN | 4 | 14.27 | >100 | >50 |
| Anacardic acid | Pilot | KCN | 5 | 3.35 | >100 | >50 |
| 6-heptadeca-9Z,12Z-dienyl salicylic acid | Deriv./Analog | KCN | 5 | 3.97 | – | – |
| NPD10211 | Pilot | KCN/Dev. & Viability | 6 | 14.46 | – | – |
| NPD390 | Deriv./Analog | KCN/Dev. & Viability | 6 | 16.56 | 45.27 | 3.20 |
| NP974 | Deriv./Analog | KCN/Dev. & Viability | 6 | 18.96 | – | – |
| NPL50654-01 | Pilot | KCN/Dev. & Viability | 7 | 7.81 | 29.38 | 3.20 |
| NPD5176 | Pilot | Dev. & Viability | 8 | – | 83.32 | >50 |
| NPD5219 | Pilot | Dev. & Viability | 9 | – | 38.73 | 14.84 |
| HTD0465 | Pilot | Dev. & Viability | 10 | – | 35.32 | 20.46 |
| STK418118 | Pilot | Dev. & Viability | 11 | – | 43.27 | >50 |

Dose-response data of L1 wild-type C. elegans (KCN survival and liquid development/viability assays) and HEK293 cells for hit compounds identified in C. elegans screens of RIKEN NPDepo natural product libraries. Lethal concentration 50% ($LC_{50}$) values estimated from fitted dose-response curves are shown for each respective assay; all dose-response data are the mean of at least three biological replicates. "–" indicate values that were not determined.

solid evidence of a benzimidazole compound that acts as a specific and potent inhibitor of complex I.

Our data suggest that NPD8790 specifically targets complex I in the electron transport chain but we wanted to determine if NPD8790 might also affect the beta tubulin BEN-1 which is the target of all commercial benzimidazoles[4,35–37]. We confirmed that commercially available benzimidazole anthelmintics potently kill *C. elegans* in normoxia (Fig. 4b and Supplementary Fig. 7) and that either deletion of *ben-1* or introduction of *ben-1* mutations known to cause benzimidazole resistance in parasites[36,37] greatly reduce sensitivity as expected (Fig. 4b). However, we find that *ben-1* mutation or loss has no effect on NPD8790 activity. The small effect on growth at high concentrations of NPD8790 is unaffected by *ben-1* mutation and the inhibition of RQ-dependent metabolism by NPD8790 is not affected by mutations in *ben-1* (Fig. 4b). Finally, we purified mitochondria from several *C. elegans* strains carrying *ben-1* mutations and find that mutations in *ben-1* have no effect on the ability of NPD8790 to inhibit complex I (Fig. 4d). We thus conclude that NPD8790 is a benzimidazole compound that has a distinct mode of action to all available benzimidazole anthelmintics. While commercial benzimidazole drugs target BEN-1 and not complex I, NPD8790 targets complex I and not BEN-1.

## Structure-activity analysis identifies requirements for potent complex I inhibition by NPD8790-related benzimidazole compounds

We identified NPD8790 as a benzimidazole compound that acts as a species-selective complex I inhibitor. To see if we could identify related compounds with either higher potency or higher species selectivity, we screened 1,280 structurally related benzimidazole compounds. We first assayed each compound in our RQ-dependent metabolism assay at 50 $\mu$M to identify benzimidazole compounds that potentially shared similar bioactivity with NPD8790. A total of 84 benzimidazole compounds (6.6%) reduced motility by at least 75% in our RQ-dependent metabolism assay (Fig. 5a). 51 of these 84 compounds were reordered (the remainder were very costly or unavailable) to perform more comprehensive dose responses. Each of the 51 compounds were assayed for their effect on complex I activity in both *C. elegans* and bovine mitochondria (Fig. 5b, c), as well as their ability to affect *C. elegans* growth and RQ-dependent metabolism. Finally, we also assayed toxicity in HEK293 cells. These data are all shown in Supplementary Data File 1. Encouragingly, all 51 benzimidazole hit compounds inhibited *C. elegans* complex I in vitro, supporting our hypothesis that inhibition of complex I by NPD8790 and structural analogs is the mechanism by which these compounds affect RQ-dependent metabolism (Fig. 5b and Supplementary Data File 1).

To gain insight into the structural requirements for benzimidazole-based compounds to act as complex I inhibitors we first compared the properties of the active and inactive benzimidazole compounds from our screen. In brief, active compounds tended to have lower molecular weights (287 Da vs. 316 Da), fewer hydrogen bond donors and acceptors (3.4 vs. 5.4), higher computed octanol/water coefficients (logP; 4.0 vs. 3.3), and lower topological polar surface area (TPSA; 31.9 vs. 59.2) (Supplementary Fig. 8). Together, this suggested smaller, more hydrophobic benzimidazole compounds were more likely to be active in our *C. elegans* assays, likely reflecting that these properties are beneficial for chemical accumulation in nematodes[41]. We next examined the chemical structure of the 51 active compounds and found almost all have a benzimidazole core which is linked to an aromatic ring—compounds with other groups or structures show much lower activity and this 'benzimidazole linked to an aromatic ring' is the consistent feature of all potent compounds. The key differences between the active molecules are in the precise position of the linkage to the aromatic ring, the length of the linkage, and the substitution of the aromatic ring. Compounds where the aromatic ring is linked via the nitrogen at position 1 of the benzimidazole core

tend to show more potent complex I inhibition than compounds with the linkage through position 2 (Fig. 5d). We also find a complex effect on the length of the linker between the benzimidazole and the aromatic group on potency: compounds with either a 1-atom linker or 3 or more atom linkers tend to be strong inhibitors but compounds with a 2-atom linker show much reduced potency (Fig. 5e).

We combined the structure-activity data in Fig. 5 to define 3 distinct structural classes of benzimidazole complex I inhibitors—we will refer to these as classes A-C (Fig. 5f). All of them have a benzimidazole core which is linked to an aromatic ring. The key differences between the subclasses are the position and the length of the linkage (Supplementary Data File 2). Class A compounds include NPD8790 and in this class the aromatic ring is linked via position 2 of the benzimidazole group via a short linkage (1-atom linkage). These compounds are good complex I inhibitors with $IC_{50}$ values well below 10 $\mu$M. More importantly, Class A includes the compounds with the strongest species selectivity e.g., NPD8790 has a >10-fold differences in $IC_{50}$ between *C. elegans* and bovine complex I (Fig. 5c). They have very little detectable effect on growth in normoxic conditions and NPD8790 also shows low toxicity against mammalian cells even at high concentrations (Supplementary Data File 2). Thus, these are good candidates as anthelmintics.

The other two classes both contain multiple compounds with substantially more potent complex I inhibition than Class A e.g., compound 5567167, a class B compound, has a 75-fold lower $IC_{50}$ for *C. elegans* complex I than NPD8790 (Fig. 5g). These compounds all have the aromatic rings linked through the 1st position of the benzimidazole core with linkages of varying lengths—the differences between the two classes is in the length of the linkage between the aromatic group and the benzimidazole core. Class B comprises compounds with a 1-carbon linkage—these are very potent inhibitors with reasonable species selectivity. Class C comprises 5-carbon linkage compounds—these show excellent potency with good species selectivity. Additionally, representatives of both Class B (STK697993) and Class C (7732524) similarly inhibit complex I from several free-living Clade V nematode species (*C. briggsae*, *P. pacificus*, and *P. hermaphrodita*) (Supplementary Fig. 9). We note that Class B and C compounds are good anthelmintic candidates but that both sets show effects on growth in normoxia unlike Class A compounds. Although they also show a slightly reduced species selectivity, many Class B and C compounds still demonstrate low toxicity against mammalian cells. Taken together, these data suggest that Class A compounds may selectively affect worms when they rely on RQ-dependent metabolism, making them the best candidates for anthelmintics.

To support complex I inhibition as the mode of action for NPD8790 analogs in our RQ-dependent metabolism assay, we next examined the relationship between the in vitro and in vivo potencies of representative compounds from our three classes (A-C) of putative benzimidazole complex I inhibitors. For comparison, we included two known complex I inhibitors (rotenone and fenazaquin) and an existing benzimidazole anthelmintic (febendazole) with no previously reported activity against complex I. We found a strong positive correlation (Pearson's correlation coefficient = 0.94) existed between the in vitro complex I $IC_{50}$ and in vivo RQ-dependent metabolism $LC_{50}$ values, supporting the argument that complex I inhibition underlies activity in our RQ-dependent metabolism assay (Fig. 6). We found similar moderate-strong correlations when we expanded our analysis to include all compounds of classes A-C for both RQ-dependent metabolism and the *C. elegans* development and viability assays (Supplementary Fig. 10), providing further evidence that complex I is the in vivo target of the NPD8790 family. While there were some differences between in vitro and in vivo activities, we suggest these likely arise because of differences in drug metabolism or uptake among analogs (e.g., cuticle permeability, drug metabolism, or efflux). Finally,

**Table 2 | Effect of RIKEN NPDs on the enzymatic activity of ETC complexes**

| % Activity | | | | | |
|---|---|---|---|---|---|
| Compound ID | Structural Group | Complex I | Complex II | Complex III | Complex IV |
| Rotenone | Control | 0.00 (0.0) | 105.97 (3.17) | 99.20 (1.32) | 99.57 (1.12) |
| Wact-11 | Control | 84.83 (1.45) | 24.43 (0.62) | 90.97 (1.75) | 89.00 (0.63) |
| Antimycin A | Control | 117.05 (0.43) | 79.54 (5.98) | 0.00 (0.0) | 94.00 (1.10) |
| KCN | Control | 128.00 (1.32) | 104.53 (5.34) | 104.75 (2.82) | 0.40 (0.20) |
| NPD8902 | 1 | 27.82 (0.70) | 117.39 (3.63) | 74.47 (2.75) | 0.84 (0.45) |
| NPD6380 | 2 | 86.43 (2.27) | 102.27 (2.46) | 107.06 (1.36) | 115.38 (4.42) |
| NPD8034 | 2 | 99.30 (2.57) | 89.02 (4.77) | 95.08 (1.51) | 96.04 (2.78) |
| NPD1450 | 2 | 84.47 (2.03) | 103.01 (4.34) | 111.72 (0.94) | 117.12 (3.30) |
| NPD8366 | 3 | 102.66 (1.57) | 86.41 (3.37) | 76.74 (1.10) | 97.72 (2.77) |
| NPD8298 | 3 | 79.90 (1.23) | 110.37 (5.08) | 101.94 (2.48) | 108.05 (2.17) |
| NPD6240 | 3 | 102.09 (1.80) | 87.73 (3.81) | 58.21 (2.30) | 26.82 (0.75) |
| FSL0005 | 3 | 91.90 (0.78) | 80.91 (2.02) | 103.92 (1.25) | 92.29 (3.01) |
| NPD10504 | 4 | 4.22 (1.25) | 106.07 (5.30) | 79.51 (1.51) | 74.57 (1.35) |
| NPD3577 | 4 | 36.82 (2.15) | 115.11 (4.34) | 65.54 (2.59) | 81.06 (2.34) |
| NPD6303 | 4 | 12.74 (0.64) | 101.47 (3.95) | 78.10 (1.39) | 83.07 (1.76) |
| NPD8790 | 4 | 5.34 (1.47) | 99.71 (4.32) | 63.54 (0.90) | 79.84 (1.34) |
| Anacardic acid | 5 | 92.95 (1.67) | 87.62 (1.34) | 0.00 (0.00) | 15.80 (1.15) |
| NPD390 | 6 | 18.50 (0.92) | 82.90 (1.87) | 85.20 (1.85) | 92.13 (2.10) |
| NPL50654-01 | 7 | 0.51 (0.26) | 82.64 (3.27) | 34.10 (0.32) | 22.84 (1.17) |

In vitro percent activity of each of the four ETC complexes in response to treatment with 100 µM of NPD hit compounds. Percent activity of each complex in response to inhibitors was calculated relative to the corresponding solvent controls. Data are the mean of at least three biological replicates; errors represent SEM.

to confirm the activity of the NPD8790 family was distinct from existing anthelmintic drugs, we tested the activity of several NPD8790 analogs against the viability of a panel of *C. elegans* anthelmintic-resistant strains (e.g., benzimidazoles[35], macrocyclic lactones[42], imidazothiazoles[43,44], Cry proteins[45], cyclooctadepsipeptides[46], ethyl benzamides[24], and amino-acetonitrile derivatives[47]). None of the existing anthelmintic-resistant strains showed any evidence of reduced susceptibility to the NPD8790 analogs, highlighting the distinct bioactivity of the NPD8790 family and their potential for tackling anthelmintic resistance (Supplementary Fig. 7c and Supplementary Table 2).

**Benzimidazole complex I inhibitors affect *H. polygyrus*, a mouse soil-transmitted helminth**

We identified a family of benzimidazole compounds that act as species-selective inhibitors of complex I. Several of these only kill *C. elegans* when they require RQ for their survival and thus have potential as anthelmintics since they may likewise kill STHs in the anaerobic conditions that they encounter in the host. To test their effect on STHs, we used *H. polygyrus*, a mouse STH parasite that has a gut-dwelling adult phase that is used widely as an STH model[48,49]. We tested 11 of our compounds covering representatives from Class A, B, and C structures on two life-cycle stages: L3 and adults. In STHs, there is a strong shift in UQ to RQ ratios between free living stages, like the L3s we test here, and adults that reside in low oxygen host niches. L3s contain high UQ and low RQ levels and thus rely on UQ-dependent aerobic metabolism. In adults the situation is reversed—they have high RQ levels and use anaerobic RQ-dependent metabolism. The switch from UQ to RQ synthesis is largely regulated by alternative splicing of the critical polyprenyltransferase *coq-2*[19]. Adults typically express high levels of the *coq-2e* isoform that is required for RQ synthesis and consistent with this, *H. polygyrus* adults express mostly the *coq-2e* isoform (67%)[50]. This shift from high UQ to high RQ allows us to compare the effect of our compounds on parasite life stages that rely mostly on UQ aerobic metabolism (L3s) with those that rely mainly on RQ (adults). Note that

during the normal life cycle, adults only live in anaerobic conditions. In the in vitro assays, however, they encounter a normoxic environment but importantly since the majority of their quinone is RQ, they still use RQ-dependent metabolism.

We find that none of our compounds showed significant activity on L3s in vitro. This weak effect on *H. polygyrus* when they are using UQ-dependent aerobic metabolism is consistent with what we find in *C. elegans*—all of our compounds are more potent when the animals rely on anaerobic metabolism. However, 5 compounds (NPD8790, STK951902, 7925515, 7925804, 7732524) have potent effects on adult viability after 24-h exposure (Fig. 7). These comprise 2/3 Class A and 3/3 Class C compounds tested. This suggests that the active compounds specifically affect *H. polygyrus* adults—this is the stage when they primarily synthesize and use RQ rather than UQ as an electron carrier. We also tested all 11 compounds on adults in anaerobic conditions using an anaerobic chamber. These conditions should partly mimic the low oxygen situation in the host and thus should prevent any residual UQ-dependent aerobic metabolism that might occur in adults, despite their low UQ levels. We note that we did not use anaerobic conditions in *C. elegans* since we could not fit our imaging equipment into an available hood, and did not use KCN for the *H. polygyrus* experiments since in vitro the *H. polygyrus* animals were unhealthy in KCN alone. We find that the anaerobic environment generally increases the activity of Class A, B, and C compounds (Supplementary Fig. 11). The effect is small, however, and we note that the adults are weakly affected by the anaerobic conditions alone as the controls show a small decrease in viability. We suggest that the limited additional effect of the anaerobic conditions is because *H. polygyrus* adults primarily make RQ and thus principally depend on RQ-dependent metabolism in both normoxic and anaerobic conditions. These in vitro experiments suggest that this family of benzimidazole compounds target Complex I in a species-selective manner and kill adult *H. polygyrus* when they rely on RQ as the electron carrier driving energy metabolism. We conclude that these compounds, or related derivatives, may have potential as anthelmintics.

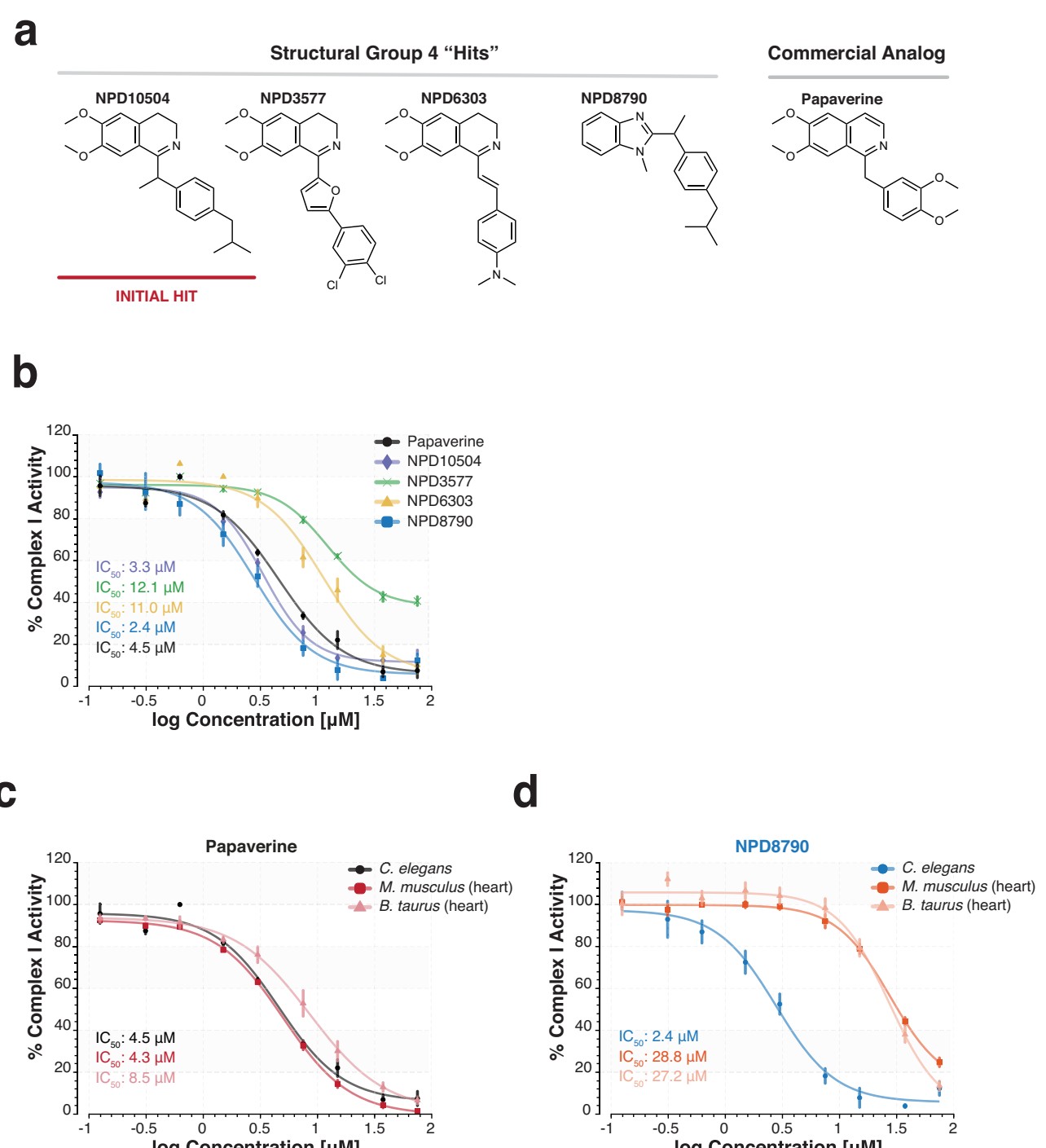

**Fig. 3 | Compounds of structural group #4 inhibit *C. elegans* mitochondrial NADH:ubiquinone oxidoreductase (complex I) in vitro. a** Chemical structures of the four molecules of structural group #4 (NPD10504, NPD3577, NPD6303, and NPD8790) and papaverine, a structurally related commercial analog. **b** Dose-response curves of papaverine and group #4 compounds against the in vitro complex I activity from wild-type *C. elegans* (N2) mitochondria. The half-maximal inhibitory concentrations or IC₅₀ values estimated from fitted curves are displayed for each compound. **c** & **d** Dose-response curves for papaverine (**c**) and NPD8790 (**d**) against in vitro complex I activity from *C. elegans*, *M. musculus* heart, and *B. taurus* heart mitochondria. IC₅₀ values estimated from fitted curves are displayed for each combination of compound and mitochondria. All data are the mean of at least three biological replicates; error bars represent SEM.

Finally, we tested 4 compounds for anthelmintic effects in vivo. We chose the two most potent compounds from Classes A and C since these classes showed the strongest effects on adults in vitro and tested them at a single high 200 mg/kg dose in mice infected with *H. polygyrus* (see Methods for details). The mice showed no toxicity following treatment with our benzimidazole compounds, however, reductions in *H. polygyrus* worm burdens relative to untreated controls (-19–41% reduction) (Table 3) were observed for all mice treated with the 4 NPD8790 analogs. Although the effects of the NPD8790 analogs were weaker than that of existing commercial anthelmintics in vivo (for example in previous studies by the Keiser group, 90%, 99%, and 62% reductions in *H. polygyrus* worm burdens were observed after a single dose of either 100 mg/kg fenbendazole, 1 mg/kg moxidectin, or 1.25 mg/kg levamisole, respectively[48,51]), we note that strong in vivo

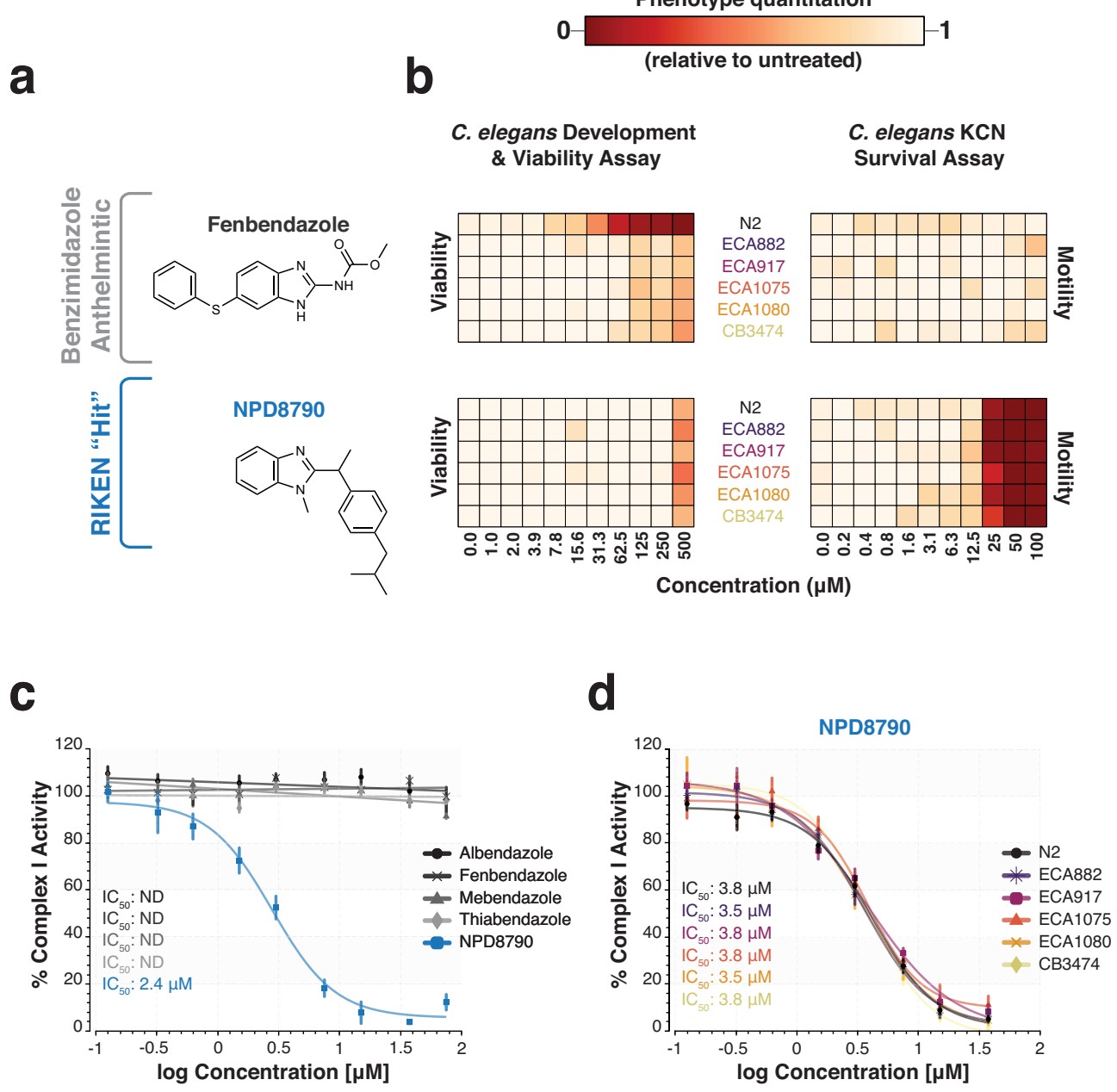

**Fig. 4 | NPD8790 shows distinct bioactivity from that of existing benzimidazole anthelmintics. a** Chemical structures of NPD8790 and the commercial benzimidazole anthelmintic, fenbendazole. **b** Dose response of fenbendazole and NPD8790 in two assays of *C. elegans* viability for six different *C. elegans* strains: N2, wild type; ECA882, deletion (*ben-1*); ECA917, F200Y (*ben-1*); ECA1075, F167Y (*ben-1*); ECA1080, E198A (*ben-1*); CB3474, G104D (*ben-1*). The color-coded scale denotes the phenotypic outcome of drug treatment (e.g., motility or viability) relative to DMSO controls; white/pale-yellow indicates *C. elegans* growth and viability similar to DMSO controls and red indicates *C. elegans* death and arrested development. **c** Dose-response curves of four commercial benzimidazole anthelmintics and

NPD8790 against the in vitro complex I activity from wild-type *C. elegans* (N2) mitochondria. $IC_{50}$ values estimated from fitted curves are displayed for each compound; "ND" indicates an $IC_{50}$ values was not determined. **d** Dose-response curves of NPD8790 against the in vitro activity of complex I from wild-type *C. elegans* (N2) mitochondria, and mitochondria from *C. elegans* benzimidazole-resistant mutant strains: ECA882, ECA917, ECA1075, ECA1080 and CB3474. NPD8790 $IC_{50}$ values estimated from fitted curves are displayed for each strain. All data are the mean of at least three biological replicates; error bars in mitochondrial assays represent SEM.

anthelmintic activity is a high standard to achieve, especially from a single oral dose. Thus we believe that the detectable in vivo activity from this preliminary testing of the NPD8790 family is an encouraging starting point. Weaker in vivo effects could result from myriad causes from rapid metabolism of the compounds by the host or gut microbiota to rapid uptake and excretion by the host and thus low levels of uptake into the parasite. We hypothesize multiple doses or longer treatment time might result in a more potent response as an inhibitor

of metabolism might take longer to have an effect than a paralytic like levamisole. We note that in vitro, the effects of our benzimidazole compounds on adult *H. polygyrus* were greater after 72 h of treatment than after only 24 h treatment (Fig. 7) which supports the idea that these may be slower-acting antiparasitic drugs. We also note that mebendazole, the front line anthelmintic for human STH infections shows relatively weak efficacy in the *H. polygyrus* murine model (63.5% reduction in worm burden for single 300 mg/kg dose)[52], suggesting

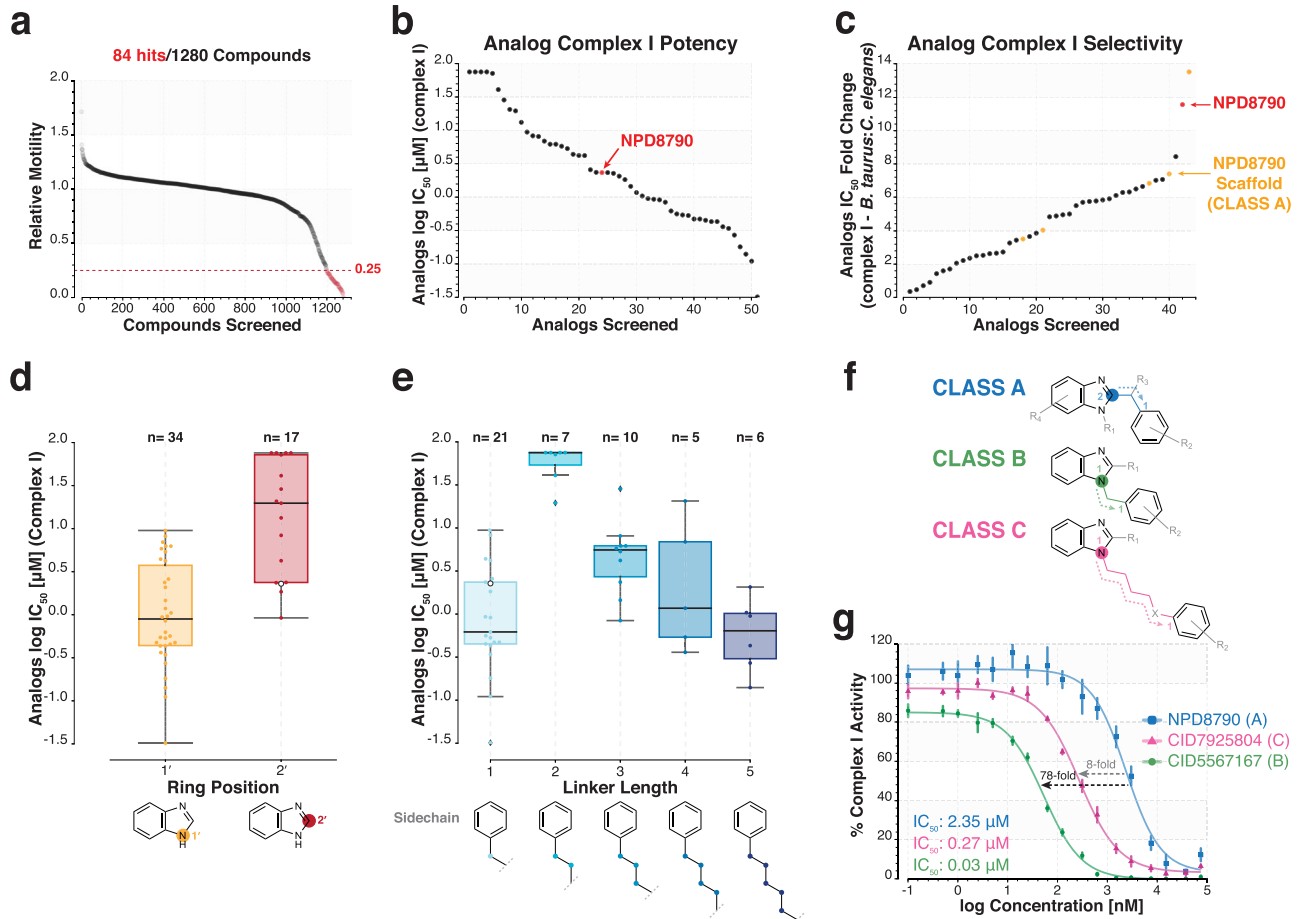

**Fig. 5 | A SAR analysis reveals potent NPD8790 analogs with similar bioactivity.** **a** Screening results of 1,280 benzimidazole-containing compounds (50 μM) against L1 wild-type *C. elegans* in the KCN survival assay (see Methods). Survival and overall health of worms was assessed relative to DMSO controls by monitoring worm motility 3 h after the removal of KCN. Data were sorted by relative motility score and hits were identified as compounds that reduced motility below 0.25; 84 hits were identified. **b** Log IC₅₀ values of 51 NPD8790 analogs against the in vitro activity of complex I from wild-type *C. elegans* (N2) mitochondria; compounds are arranged in ascending order of potency (IC₅₀); the position of NPD8790 is highlighted in red. **c** Fold-change in IC₅₀ between the in vitro activity of *B. taurus* and *C. elegans* complex I for 43 NPD8790 analogs; compounds are arranged in ascending order of *C. elegans* selectivity; NPD8790 and related scaffolds are highlighted in red and orange, respectively. **d** and **e** Comparison of the inhibitory potency (IC₅₀ values) of NPD8790 analogs against *C. elegans* complex I when subdivided on chemical structure: (**d**) the benzimidazole core position from which tail moieties originate (i.e., 1′ or 2′) or (**e**) the number of linker atoms in the tail moiety (i.e., 1, 2, 3, 4, 5). The position of NPD8790 is highlighted in white; analogs with either 5′ sidechain substitutions (n = 0) or sidechain lengths of 0 (n = 1) or 10 (n = 1) were insufficient for comparison. Within each boxplot, black horizontal lines denote group medians, boxes span from the 25th to the 75th percentile of each group's distribution, and whiskers extend to the maximum and minimum values. **f** Chemical structures of class A, B and C benzimidazole analogs. **g** Dose-response curves for NPD8790 (Class A) and two analogs (5567167, Class B; 7925804, Class C) against the in vitro activity of complex I from wild-type *C. elegans* (N2) mitochondria. IC₅₀ values estimated from fitted curves and fold-changes in potency among NPD8790, 5567167 and 7925804 are highlighted on the plot. Data are the mean of at least three biological replicates; error bars represent SEM.

other STH species may similarly have increased susceptibility to the NPD8790 family. We suggest that medicinal chemistry studies will be needed to identify derivatives that increase the stability or delivery of these benzimidazole-based complex I inhibitors to the parasites in vivo.

## Discussion

Soil-transmitted helminths (STHs) are major human pathogens. While there are excellent frontline anthelmintics including macrocyclic lactones and benzimidazoles, the number of classes of anthelmintics is very limited and resistance is increasing and is already very high for some livestock parasites. There is thus an urgent need for new classes of anthelmintic drugs. In this study we focused on using *C. elegans* to screen for drugs that specifically inhibit the unusual anaerobic metabolism that STHs like *Ascaris* and hookworm use to survive the low oxygen conditions in the host gut. This metabolism requires rhodoquinone (RQ), an electron carrier that is absent from the hosts.

We screened a library of natural products and their derivatives (NPDs) that contained both previously characterized and uncharacterized compounds and identified multiple distinct structural families of compounds that kill *C. elegans* specifically when they are surviving using RQ-dependent metabolism. These are candidate anthelmintics since they may similarly kill STHs in vivo when they rely on RQ-dependent metabolism for survival in the gut.

During RQ-dependent metabolism, electrons enter the ETC solely through complex I and exit through complex II and consistent with this we had previously shown that known complex I inhibitors and complex II inhibitors are potent inhibitors of RQ-dependent metabolism. We identify both specific complex I and complex II inhibitors in our screens—a distinct set of benzimidazole compounds as complex I inhibitors and siccanin as a complex II inhibitor. The way these two sets of compounds affect RQ-dependent metabolism is consistent with our previous findings as these are expected to block electron entry or exit from the ETC. Other RQ-dependent metabolism hits are more

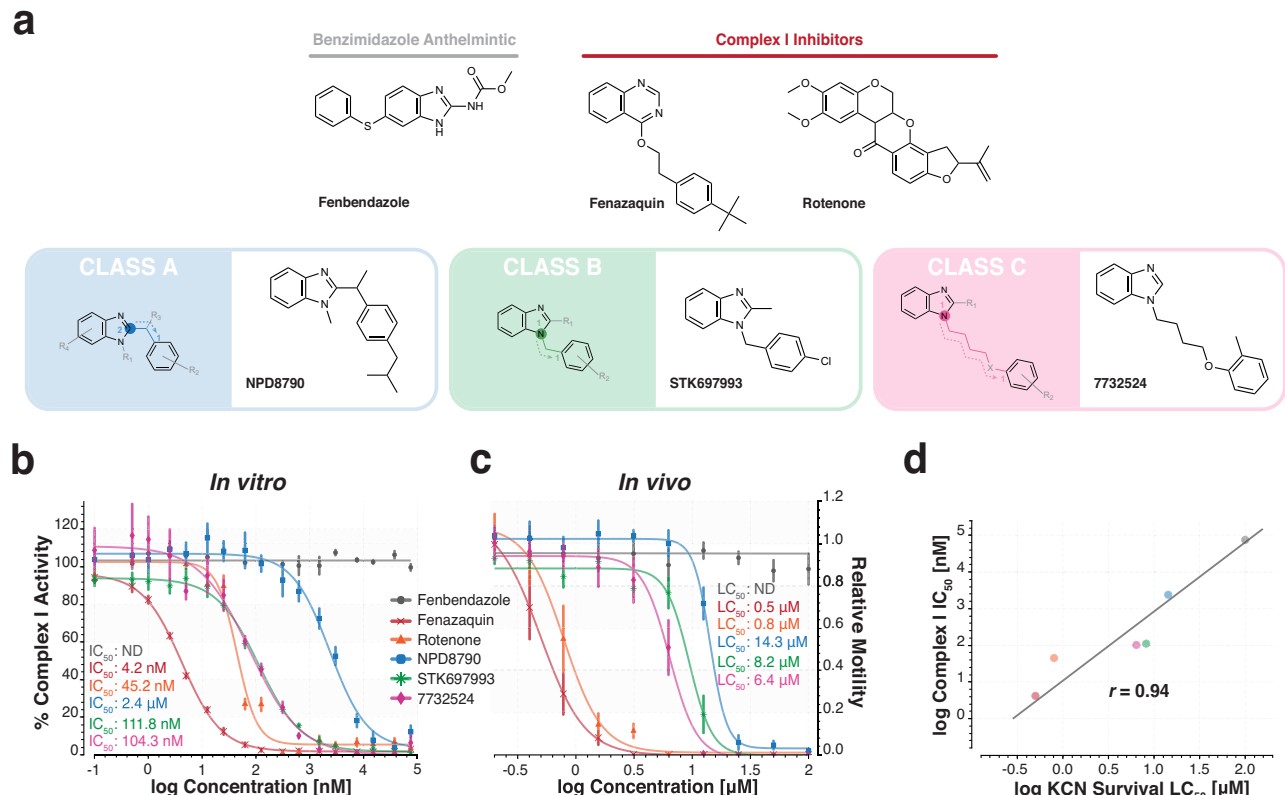

**Fig. 6 | Chemical inhibition of complex I activity in vitro is correlated with lethal activity in the _C. elegans_ RQ-dependent metabolism assay in vivo.**
**a** Chemical structures of class A (NPD8790), B (STK697993) and C (7732524) benzimidazole analogs; rotenone and fenazaquin, known complex I inhibitors; and fenbendazole, a benzimidazole anthelmintic. **b** Dose-response curves of each of the six compounds against the in vitro activity of complex I from wild-type _C. elegans_ (N2) mitochondria. **c** Dose-response curves for each of the six compounds against wild-type L1 _C. elegans_ in the RQ-dependent KCN survival assay. $IC_{50}$ (**b**) and $LC_{50}$ (**c**) values estimated from dose-response curves in each respective assay are shown for each of the six compounds. All data are the mean of at least three biological replicates; error bars represent SEM. **d** In vivo $LC_{50}$ values (**c**) from the KCN survival assay were plotted against the in vitro $IC_{50}$ values (**b**) from complex I enzymatic assay for each compound. A Pearson's correlation coefficient of 0.94 (strong correlation) was calculated for the relationship between in vitro and in vivo activities.

enigmatic however. For example, in our hands anacardic acid is a potent inhibitor of complexes III and IV in purified mitochondria (Table 2) and it is unclear mechanistically how it might be acting or why it has such a strong effect on RQ-dependent metabolism but appears to have no detectable effect on metabolism in normoxia. Other RQ-dependent metabolism hits appear to have no effect on the ETC and must be affecting RQ-dependent metabolism in some other unknown manner, perhaps through targeting other required components of RQ-dependent metabolism such as AMPK[53] or RQ synthesis itself. Thus while the mode of action of some of the RQ-dependent metabolism hits fits with what is known of electron flow in the ETC in RQ-dependent metabolism, others will require future studies to identify the targets and mode of action.

The main focus of our analysis was on a distinct, previously uncharacterized family of benzimidazole compounds that we identified as potent and specific inhibitors of mitochondrial respiratory complex I. Benzimidazoles like mebendazole and albendazole are well characterized anthelmintics that target BEN-1, a beta tubulin. The benzimidazole compounds we identify have a distinct mode of action: we showed that their activity is not affected by mutations in _ben-1_, and that known benzimidazole anthelmintics do not affect complex I activity. This confirms that the inhibition of complex I by this family of compounds is a distinct activity for benzimidazoles. We note that there is a single previous report of a compound with insecticidal activity and a benzimidazole core that is described as a putative complex I inhibitor (e.g., 1-(3,7-dimethyl-7-ethoxy-2-octenyl)−2-methylbenzimidazole)[54]. However, the structure is distinct to the hits we have identified here—

instead of an aromatic group linked to the benzimidazole core, it has a long terpenoid chain. Notably, its inhibition of complex I has only been shown in mammalian mitochondria (e.g., rat liver, bovine heart) and has not been shown for any species of nematode[54,55]. Furthermore, no species selectivity has ever been demonstrated for this compound.

We identified the complex I inhibitors in _C. elegans_ but also tested their efficacy on _H. polygyrus_, a mouse STH that has a gut-dwelling adult phase. The compounds show weaker effects on _H. polygyrus_ larvae, which contain largely UQ and thus do not use RQ-dependent metabolism. In contrast, several of the compounds act as potent anthelmintics in vitro against adult _H. polygyrus_. _H. polygyrus_ adults, like other STHs, show a shift in splicing of _coq-2_ consistent with a shift from UQ synthesis to RQ synthesis and containing mostly RQ and not UQ, and these findings are consistent with a model that these benzimidazole-based complex I inhibitors specifically kill nematodes when they rely on RQ-dependent metabolism. We tested a small number of the most potent compounds in vivo to determine their efficacy in clearing a murine _H. polygyrus_ infection. All compounds reduced _H. polygyrus_ worm burdens relative to controls, however, the effects were weaker than that observed for existing commercial anthelmintics and this could be due to a number of complex factors including rapid metabolism of the compounds by either host or host gut microbiota or low levels delivered to the parasite in vivo because of rapid uptake by the host. We delivered only a single dose in vivo but note that longer exposure to the drugs in vitro resulted in more potent killing (Fig. 7) suggesting a multidose regimen might improve in vivo efficacy. We anticipate that a phase of medicinal chemistry is needed to

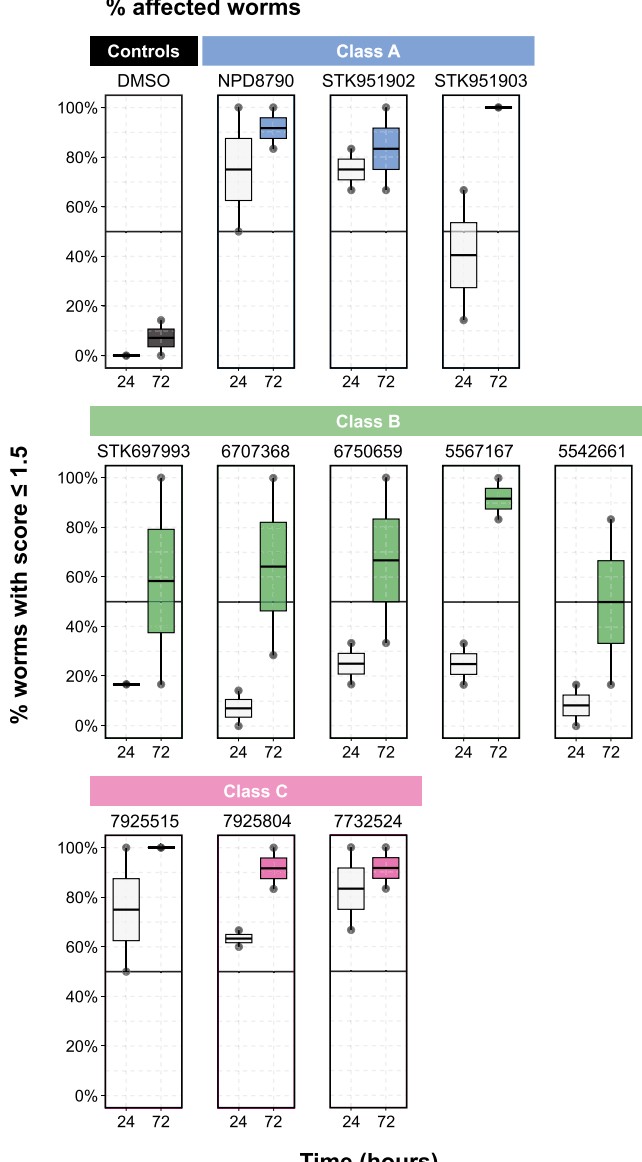

**% affected worms**

**Fig. 7 | Benzimidazole complex I inhibitors affect adult *H. polygyrus* viability in vitro.** The activities of eleven NPD8790 analogs were tested against adult *H. polygyrus* viability in vitro under aerobic conditions for 24 and 72 h (see Methods). After either 24- or 72 h chemical treatment, worm viability was manually scored by an examiner using a scale from 0 (dead) to 3 (motile); DMSO was used as a negative control in both 24- and 72 h treatments. The percentage of affected worms (i.e., worms with a score equal to or lower than 1.5) in each biological replicate is shown; data are from two biological replicates. Within each boxplot, black horizontal lines denote group medians, boxes span from the 25th to the 75th percentile of each group's distribution, and whiskers extend to the maximum and minimum values.

identify derivatives with higher in vivo efficacy and that similar lead optimization strategies have proved successful for almost all other anthelmintic compounds in the past, including the recently developed anthelmintic, monepantel[47]. Finally we note that while the murine *H. polygyrus* model is an excellent screening tool it is not a perfect one—for example the frontline human anthelmintic mebendazole shows relatively weak efficacy in a murine *H. polygyrus* model[52]. Additionally, given the large abundance of commercially available benzimidazole analogs and existing studies that have examined the bioavailability of benzimidazole anthelmintics, we are optimistic that improvements can be made to the in vivo activity of the NPD8790 family. We thus

**Table 3 | In vivo activity of NPD8790 analogs against *H. polygyrus***

| Treatment | Class | Dose (mg/kg) | # Mice cured/ # Mice investigated | Median # of living *H. polygyrus* worms | % Worm burden reduction |
|---|---|---|---|---|---|
| Control | – | – | 0/4 | 124.0 (6.0) | – |
| NPD8790 | A | 200 | 0/4 | 95.5 (4.0) | 23.0 |
| STK951902 | A | 200 | 0/4 | 73.0 (1.5) | 41.1 |
| 7925515 | C | 200 | 0/4 | 100.0 (8.0) | 19.4 |
| 7732524 | C | 200 | 0/4 | 93.5 (5.0) | 24.6 |

Worm burdens of mice infected with *H. polygyrus* were assessed 1 week after a single oral dose of 200 mg/kg of one of four different NPD8790 analogs (NPD8790, STK951902, 925515, 7732524). The median number and median absolute deviation (MAD) of *H. polygyrus* adults collected from intestine are reported for three to four mice in each drug treatment. Percent reduction in worm burden was calculated by comparing the median worm counts between control and NPD8790 analog-treated mice. Data are one biological replicate of four mice (technical replicates). "–" indicate values that were not determined.

believe the NPD8790 family of complex I inhibitors merit further drug development and have potential to lead to new anthelmintics.

In summary, we screened a natural product library in *C. elegans* to identify compounds that killed *C. elegans* only under conditions when they require RQ-dependent metabolism to survive. We believe that this is a reasonable model for helminths which rely on RQ-dependent metabolism to survive in the highly anaerobic environment of a host gut. We identified compounds that specifically inhibit either complex I or complex II in a species-specific manner. Complex I and II are the entry and exit points for electrons in the RQ-coupled ETC and thus the mode of action is consistent with what is known of the ETC in RQ-dependent metabolism. The complex I inhibitors we identified are a family of compounds with a benzimidazole core, which represents a distinct activity for benzimidazoles—well characterized benzimidazole anthelmintics have no activity against complex I and target microtubules. We thus believe that, to the best of our knowledge, we have uncovered a unique class of benzimidazoles that specifically target the unusual anaerobic metabolism of helminths and that derivatives of these lead compounds may act as anthelmintics.

## Methods

### Ethical statement

All animal experimental procedures and husbandry in this study were performed by trained personnel in accordance with either the ethical committee of the canton Basel-Stadt (permission no. 520) and the University of Basel, or the Canadian Council on Animal Care with approval from the Animal Care Committee at the University of Toronto.

### Chemical sources

Natural Product Libraries (Authentic & Pilot) and related derivatives used in preliminary screens were provided by the RIKEN Natural Product Depository (NPDepo). Identified hit compounds were purchased from Vitas-M for retesting. NPD8790 analogs were purchased from ChemBridge Corporation and Vitas-M. Wact-11 (6222549) was purchased from ChemBridge Corporation. Rotenone, antimycin A, potassium cyanide, albendazole, mebendazole, thiabendazole, and fenbendazole were purchased from MilliporeSigma. Siccanin was purchased from Cayman chemical. Chemical stock solutions were made for each compound by dissolving a weighed amount of chemical into a measured volume of sterile solvent (DMSO, ethanol, or water depending on compound solubility), stock solutions were then sterile filtered using 0.2-micron filters when appropriate. All stock solutions were stored at −20 °C prior to use apart from KCN, which was prepared fresh before each experiment.

### Free-living nematode strains and maintenance

In addition to the *Caenorhabditis elegans* wild-type N2 strain, this work included the following *C. elegans* strains: CB1003 *(kynu-1(e1003) X)*; anthelmintic-resistant strains−CB3474 *(ben-1(e1880) III)*, CB193 *(unc-29(e193) I)*, HY496 *(bre-1(ye4) IV)*, RB2119 *(acr-23(ok2804) V)*, JD608 *(avr-14(ad1302) I; avr-15(ad1051) V; glc-1 (pk54) V)*, NM1968 *(slo-1(js379) V)*, RP2635 *(mev-1(tr355) III)*; benzimidazole-resistant CRISPR strains−ECA882 *(ben-1(ean64) III)*, ECA917 *(ben-1(ean98) III)*, ECA1075 *(ben-1(ean143) III)*, ECA1080 *(ben-1(ean148) III)*. This work also featured several additional free-living nematode strains: *C. briggsae* strain AF16, *P. pacificus* strain PS312, and *P. hermaphrodita* strain B178. The four CRISPR-generated benzimidazole-resistant *C. elegans* mutants were generously provided by Prof. Erik Andersen (Johns Hopkins University); PS312 and B178 were generously provided by Prof. Peter Roy (University of Toronto); all other strains were obtained from the *Caenorhabditis* Genetics Center (CGC, University of Minnesota). All strains were cultured and propagated at 20 °C on NGM (nematode growth medium) agar plates seeded with *E. coli* OP50 (Stiernagle, 2006)[56].

### *C. elegans* KCN survival assay−time-course and dose-response experiments

KCN survival assays were performed as previously described (Del Borrello et al.)[18]. In brief, 25 μL M9 buffer was distributed to each well of a 96-well flat-bottom culture plate and 0.5 μL chemical solution (or DMSO alone) was added to designated wells (1% DMSO v/v) using either a multi-channel pipette (dose responses) or pinning tool (chemical screens) (V&P Scientific). First larval stage (L1) worms were isolated from mixed-stage plates using 96-well 11 μm nylon mesh filter plates (Millipore Multiscreen) and diluted with M9 to a concentration of ~6 worms/μL; 20 μL worm suspension was added to each well of the microplate. 5 μL freshly prepared 10 X (2 mM) potassium cyanide (KCN) stock solution was then added to each well to achieve 200 μM KCN. Following KCN addition, plates were sealed with aluminum foil and incubated for 15 h at 20 °C with shaking at 165 rpm. After 15 h, KCN in plates was diluted 6-fold with M9 buffer and imaged on a Nikon Ti Eclipse inverted microscope (Nikon NIS Elements AR) every 10 min for 3 h. At each time point, two sequential images (500 ms interval) were taken for each well. Using Python scripts, worm-associated pixels in each image were segregated from background and worm motility (i.e., change in worm-associated pixels) between sequential images was used to estimate worm survival[57]. The motility score for each well was divided by the motility scores from corresponding DMSO controls, resulting in a relative motility value for each chemical. For chemical screens and dose-response experiments, the relative motility of each well 3 h post dilution of KCN was used as an endpoint. At least three biological replicates were performed for each experiment; relative motility values were averaged across replicates.

### Free-living nematode liquid development and viability assay−dose-response experiments

Liquid development and viability assays were adapted from a previously established chemical screening protocol (Burns et al.)[24]. In brief, a saturated culture of *E. coli* HB101 was concentrated 2-fold in liquid NGM and 40 μL of this suspension was distributed to each well of a 96-well flat-bottom culture plate. 0.5 μL chemical solution (or DMSO alone) was added to each well (1% DMSO v/v) using either a multi-channel pipette or a pinning tool (V&P Scientific). L1 worms synchronized from an embryo preparation performed the day prior were diluted to a concentration of ~2 worms/μL in M9 buffer; 10 μL worm suspension was then added to each well. Culture plates were sealed with parafilm and placed in a plastic container lined with wet paper towels to prevent evaporation. Containers were then incubated for 6 days at 20 °C with shaking at 165 rpm. Following incubation, 300 μL M9 was added to each well to decrease the turbidity of the HB101-NGM

suspension. Single images of each microplate well were then taken on a Nikon Ti Eclipse inverted microscope (Nikon NIS Elements AR). Images were processed using Python scripts to isolate worm-associated pixels from background. Scores of worm development and viability were determined as the sum of worm-associated pixels in each well. Development/viability scores for each well were divided by the development /viability scores for the corresponding DMSO controls, resulting in a relative viability value for each chemical. At least three biological replicates were performed for each dose-response experiment; relative viability values were averaged across replicates.

### *C. elegans* Chemical Screens

NPD libraries were stored in 96-well plates, each containing 80 compounds and 16 DMSO controls (columns 1 and 12). For preliminary screens of NPD libraries, eight 96-well plates containing 617 compounds (400 Pilot library; 80 Authentic library; 137 derivatives) and 128 DMSO controls were screened in triplicate at a concentration of 50 μM for both *C. elegans* assays described above; data were averaged across the three biological replicates. To account for possible variation among plates, hit identification was performed on a plate-to-plate basis. As drug hits (i.e., outliers) can influence the mean of the distribution of scores, relative motility or relative viability values were converted to robust z-scores using the median and the median absolute deviation (MAD) of each plate. Hits were identified if a compound's z-score was below −3 (i.e., 3x MAD below the median).

### HEK293 cell culture and dose-response experiments

Human embryonic kidney (HEK293; Invitrogen's Flp-In-293 cell line cat #R75007) cells were grown and maintained in Dulbecco's Modified Eagle's Medium (DMEM) containing 10% fetal bovine serum (FBS) and 1% penicillin-streptomycin (PenStrep).

To prepare cells for viability testing, 5,000 HEK293 cells/100 μL were seeded into each well of a 96-well flat-bottom tissue culture plate and grown overnight at 37 °C with 5% $CO_2$. 0.5 μL compound from prepared dose-response plates were then added to each well (0.5% v/v DMSO) and cells were allowed to grow for an additional 2 days. After this period of growth, cells were incubated for 4 h with 10 μL CellTiter-Blue viability reagent. Raw scores of cell viability were then determined using a CLARIOstar Plate Reader (560/590 nm) and the fluorometric quantification of the reduction of resazurin to resorufin. "Relative viability" was determined by dividing the background-corrected scores of chemical-treated wells from the corresponding DMSO controls. Data represent the mean of at least three biological replicates.

To prepare cells for mitochondria isolation, cells were grown in 5 ×20 cm dishes containing 20 mL DMEM/10%FBS/1% PenStrep for 3–4 days until confluent at 37 °C with 5% $CO_2$. Following growth, media was removed and plates were rinsed once with 5 mL of PBS. TrypLE Express Enzyme (Thermo Fisher #12604013) was then added dropwise to the plates to completely cover the surface (~2 mL) and plates were incubated at 37 °C for 5 min in order to detach cells. Cell suspension from each plate was then transferred to 18 mL of DMEM/10%FBS/1% Pen Strep media and all samples were pooled (100 mL total volume). Cell number was then determined using a Countess Automated Cell Counter (Thermo Fisher) to ensure an appropriate number of viable cells ( ~100,000,000) was obtained. Following counting, cells were pelleted by centrifugation at 200 x g for 5 min at room temperature.

### Mitochondria isolation from free-living nematodes

Mitochondria were isolated from the following free-living nematode strains: *C. elegans*−N2, CB3474, ECA882, ECA917, ECA1075, ECA1080; *C. briggsae*−AF16; *P. pacificus*−PS312; and *P. hermaphrodita*−B178. With the exception of the *P. hermaphrodita* strain B178, adult worms of each strain were grown in bulk through liquid culture to obtain mitochondria in sufficient quantities. Due to poor growth in liquid media, mixed stage populations of *P. hermaphrodita* were washed off and

collected from NGM plates seeded with OP50. The isolation of mitochondria from all free-living nematode strains followed the protocol previously outlined in Burns et al.[24]. In brief, a 1 L saturated culture of *E. coli* HB101 was concentrated 20-fold in 50 mL complete S-medium. -500,000 L1 worms synchronized from an embryo preparation performed the day prior were added to the S-media/HB101 suspension and grown to adulthood over the course of 3.5 days at 20 °C with shaking at 165 rpm. Following growth, worms were collected in 10 ×15 mL conical tubes and washed 7 times in M9 buffer. Worm suspensions were then combined into a single 15 mL conical tube and diluted to a final volume of 15 mL with M9 buffer. 1 mL aliquots of worm suspension were then distributed to 15 × 1.5 mL screw-cap microcentrifuge tubes and were pelleted at 500 g for 1 min. Excess M9 was aspirated without disturbing the worm pellet. 600 $\mu$L ice-cold isolate buffer A (250 mM sucrose, 10 mM Tris (pH 7.5), 1 mM EDTA, 1 mM PMSF (freshly prepared)) and 300 $\mu$L cold glass beads were then added to each tube. Tubes were incubated on ice for 10 min, and the worms were then lysed by bead beating (6 x 30 s, with 1 min cooling intervals). Tubes containing worm lysate were centrifuged at 1,000 g for 10 min at 4 °C and the supernatant was transferred to 15 new microcentrifuge tubes on ice. The tubes were centrifuged a second time at 1,000 g for 10 min at 4 °C and the supernatant was once again transferred to 15 new microcentrifuge tubes on ice. These tubes were then centrifuged at 16,000 g for 10 min at 4 °C. The resulting mitochondrial pellet was washed twice by re-suspension in 300 $\mu$L of ice-cold isolate buffer B (250 mM sucrose, 10 mM Tris (pH 7.5), 1 mM EDTA). Following the final wash, isolation buffer was aspirated from each tube and the pellets were flash-frozen in ethanol/dry ice and stored at −80 °C until needed.

### Mitochondria isolation from mammalian tissue

Six 8–10-week-old C57Bl/6 female mice (Charles River) were freshly dissected, and their livers and hearts collected into separate 1.5 mL microcentrifuge tubes. Organs were flash-frozen in liquid nitrogen and stored at −80 °C prior to use. The heart of a freshly slaughtered adult cow was obtained from a local abattoir (Peel Sausage Inc.), chopped into ~1-inch cubes and collected into 50 mL falcon tubes. Falcon tubes were flash-frozen in a container of dry ice and stored at −80 °C prior to use. Isolation of mitochondria from all mammalian tissue was carried out following the protocol described above. However, due to size, the initial homogenization of tissue was performed by 15 strokes in a dounce homogenizer (mouse heart) or with several 5 s pulses in small Nutribullet blender (cow heart). All mouse protocols for mouse care and dissection were reviewed and approved by the University of Toronto Animal Care Committee, in accordance with the Canadian Council on Animal Care.

### Mitochondria isolation from HEK293 cells

Isolation of mitochondria from HEK293 cells was carried out following the same protocol as the isolation of mitochondria from mammalian tissue (described above) with one exception. The initial homogenization of the pelleted cells was performed with 20 up-and-down passes of a Teflon/glass Potter-Elvehjem homogenizer with a 0.1-0.15 mm clearance. Subsequent steps were performed as previously described in Burns et al.[24].

### Electron transport chain (ETC) assays

The enzymatic activities of individual ETC complexes (I-IV) were assessed spectrophotometrically in isolated mitochondria using a Varioskan LUX Multimode plate reader (ThermoFisher Scientific). Mitochondrial pellets previously isolated were thawed on ice and resuspended in ice-cold isolation buffer (250 mM sucrose, 10 mM Tris (pH 7.5), 1 mM EDTA). The BCA assay (Walker, 1994) was used to quantify the protein concentration of mitochondria suspensions; solutions were diluted to a concentration of 0.2 mg/mL for use in all assays. For the complex I assay, mitochondria pellets were freeze-

thawed on ice three times to increase enzyme accessibility. To set up assays, chemical solutions or solvent controls (2.4% solvent v/v) were prepared in 180 $\mu$L respective enzyme buffer (see recipes below) and added to each well of a 96-well flat-bottom tissue culture plate. 20 $\mu$L mitochondria suspension was then added/mixed with a pipette to initiate the reaction. Immediately following the addition of mitochondria, absorbance was measured (complex I and II: 600 nm; complex III and IV: 550 nm) in 30 s intervals over 15 min. Enzymatic activity was determined in each reaction by plotting absorbance versus time and calculating the slope of the curve during the initial linear phase of the reaction. Any non-specific enzymatic activity was removed by subtracting the slope from corresponding control wells containing enzyme-saturating doses of inhibitor (complex I: 10 $\mu$M rotenone; complex II: 100 mM malonate; complex III: 10 $\mu$M antimycin A; complex IV: 300 $\mu$M KCN). Percent activity of each complex was calculated by dividing the enzyme-specific activity of each well by the enzyme-specific activity of the corresponding solvent controls.

Rotenone-sensitive complex I (NADH: decylubiquinone oxidoreductase) activity was assessed using the 2,6-dichlorophenolindophenol (DCIP)-coupled method previously optimized in Long et al.[58]. Chemical solutions were prepared in complex I assay buffer composed of 25 mM KPi buffer (pH 7.5), 3 mg/mL bovine serum albumin (BSA), 80 $\mu$M NADH, 60 $\mu$M decylubiquinone, 160 $\mu$M DCIP, 2 $\mu$M antimycin A and 2 mM KCN.

Complex II (succinate dehydrogenase) activity was assessed using the DCIP-coupled method previously described in Burns et al.[24]. Chemical solutions were prepared in complex II assay buffer composed of 1X PBS, 0.35% BSA, 20 mM succinate, 240 $\mu$M KCN, 60 $\mu$M DCIP, 70 $\mu$M decylubiquinone, 25 $\mu$M antimycin A, 2 $\mu$M rotenone.

Antimycin A-sensitive complex III (decylubiquinol-cytochrome C deductase) activity was determined by measuring the reduction of cytochrome C (CytC) as optimized in Luo et al.[59]. Decylubiquinol for complex III assays was prepared fresh as described in Janssen and Boyle, 2019[60]. In brief, several flakes of potassium borohydride were mixed into 10 mM decylubiquinone dissolved in ethanol, 0.1 M HCl was then added in 5 $\mu$L increments until the solution turned colorless. The solution was spun down at 10,000 g for 1 min to pellet potassium borohydride, and decylubiquinol was transferred to a fresh tube. Chemical solutions were prepared in complex III assay buffer composed of 50 mM Tris-HCl (pH 7.5), 4 mM NaN$_3$, 50 $\mu$M decylubiquinol, 50 $\mu$M oxidized CytC, 0.01% BSA, and 0.05% Tween-20.

Complex IV (cytochrome C oxidase) activity was determined by following the oxidation of CytC in the protocol outlined by Janssen and Boyle, 2019[60]. Reduced CytC was prepared fresh by adding 1 $\mu$L increments of 1 M dithiothreitol to a 1 mM solution of CytC and vortexing. A colour change from brown to orange/pink signals when CytC has been reduced. Reduction of CytC was checked by diluting a sample 50-fold and measuring the ratio of absorbance 550/560 nm (ratio > 6 indicates reduction). Chemical solutions were prepared in complex IV assay buffer composed of 50 mM KPi buffer, 60 $\mu$M reduced CytC.

For preliminary screens of hit compounds, compounds were screened at 100 $\mu$M; data are the average of at least three biological replicates. The complex I and II activity assays were used for additional dose-response experiments. Assays were performed to cover chemical concentrations ranging from 0.1 nM to 75 $\mu$M. For dose responses of benzimidazole analogs, data represent the average of two biological replicates. For all other dose-responses, data represent the average of at least three biological replicates.

### Animals

The life cycle of *H. polygyrus* is maintained at the Swiss Tropical and Public Health Institute (Swiss TPH). The ethical committee of the canton Basel-Stadt authorized all performed animal experiments (permission no. 520). Only trained personnel could execute the experiments.

For in vivo testing, female 3-week old NMRI mice were purchased from Charles River, Germany. The mice were kept in individually

ventilated cages (IVC) under environmentally controlled conditions (temperature: 22 °C ± 2 °C; relative humidity: 55% ± 15%; artificial lighting with a circadian cycle of 12 h of light) and had free access to water (municipal tap water supply) and rodent food. Rodents were allowed to acclimatize to the new environment for 1 week before infection. Upon 2 days of arrival, dexamethasone (0.25 mg/L) (Sigma-Aldrich, Switzerland) was added to the normal drinking water for immunosuppression to enable parasite establishment. After 1 week, mice were orally infected with 100 *H. polygyrus* third larval stage (L3) in 150 μL water.

### Aerobic in vitro drug screening in larval and adult *H. polygyrus*

The eleven benzimidazole derivatives were tested at a concentration of 10 μM under aerobic conditions against L3 and adult *H. polygyrus*.

For the larval assay, mice feces were collected 2 weeks post-infection and eggs were isolated. The eggs were left on agar at room temperature in the dark for eight to 10 days to allow them to develop into L3. Next, 30–40 L3 were incubated (37 °C, 5% $CO_2$) in the culture medium (RPMI 1640 medium + 1% pen./strep. + 5% amphotericin B) containing the derivatives in 96-well plate (Corning, USA) reaching a final volume of 250 μL for 24 h. For the adult drug screening assay, female NMRI mice were euthanized with $CO_2$ and adult *H. polygyrus* were dissected out of the intestines. Subsequently, three worms were exposed to the culture medium (RPMI 1640 medium + 1% pen./strep. + 5% amphotericin B) with the derivatives (2 mL) in a 24-well plate (Corning, USA) for either 24 or 72 h. DMSO (<1%) served as a negative control in both assays. All conditions were tested in duplicates. Finally, 80 μL of hot water (~80 °C) for L3 and 500 μL of hot water (~80 °C) for adults were added to the wells to check for their viability under a microscope (Carl Zeiss, Germany). Larvae were categorized as being either alive or dead, depending on whether they moved or not. The adult worms were scored under a microscope based on changes in motility, viability, and morphological alterations at a scale ranging from 0–3 in steps of 0.25 (3 = motile, no changes in morphology or transparency; 2 = reduced motility and/or some damage to tegument noted, as a well as reduced transparency and increased granularity; 1 = severe reduction of motility and/or damage to tegument observed with high opacity and granularity; 0 = dead). The whole experiment was performed two times i.e., two independent biological replicates for both life stages (L3s and adults) and exposure times (24 and 72 h).

### Anaerobic in vitro drug screening in adult *H. polygyrus*

To investigate the impact of the eleven benzimidazole derivatives under anaerobic conditions, an anaerobic chamber with an integrated incubator (Coy Laboratory Products, USA) was used. The same procedure was followed as described for the aerobic drug screening in adult *H. polygyrus*, but the incubation took place within the anaerobic chamber (37 °C, 85% $N_2$, 10% $CO_2$, 5% $H_2$). Benzimidazole derivatives were tested at a concentration of 10 μM. For scoring, the plates had to be removed from the anaerobic chamber. The whole experiment was performed two times i.e., two independent biological replicates.

### In vivo efficacy studies in *H. polygyrus* mouse model

Four benzimidazole derivatives were tested for in vivo efficacy. The administration of dexamethasone was stopped after 12 days post-infection and 2 days later, the selected four benzimidazole derivatives were administered by oral gavage at a single dose of 200 mg/kg to groups of each four mice. Four mice were left untreated and served as a control group. The derivatives were prepared as a Tween80 (Sigma-Aldrich, Switzerland)/ethanol (Merck, Germany) mixture dissolved in tap water (7:3:90, v/v). After 6–7 days post-dosing, the mice were euthanized with $CO_2$ and dissected. Adult worms were removed from the intestine and counted. The worm burden reduction was calculated by comparing the number of worms found in the control animals versus the treated animals.

### Cheminformatics

Chemical structures as supplied by the manufacturers (e.g., RIKEN NPDepo, Vitas-M, ChemBridge Corporation) were analyzed in Python (v. 3.7.4) using the ScoPy library[61]. Specifically, molecular mass, number of atoms and heteroatoms, number of hydrogen bond donors and acceptors, logP (hydrophobicity), logSw (solubility), and topological polar surface area (TPSA) were computed using the default parameters.

Structural groups of natural product derivatives in the Pilot library were determined by the RIKEN natural product depository. Murcko scaffolds (for compounds with ring systems > 1) were generated for each molecule using DataWarrior (v. 5.5.0), FP2 fingerprints were then generated for each scaffold in Python (v. 3.7.4) using Pybel (v. 3.0.1)[62] and pairwise Tanimoto coefficients between fingerprints were used to assess chemical similarity. Scaffolds that shared a Tanimoto coefficient of 0.55 or greater were grouped together. Networks of similar scaffolds within each group of hit compounds in our screen are shown using Cytoscape (v. 3.8.0)[48,63].

### Statistical analyses

For all dose-response experiments, dose-response curves were fitted using non-linear regression using the DRC package (v. 2.9.4) in R (v. 3.6.1) and the Python rpy2 interface (v. 2.9.4). Inhibitory and lethality concentration 50% values (i.e., $IC_{50}$ or $LC_{50}$) parameters were estimated from the fitted curves.

### Inclusion and ethics

All authors/collaborators of this study have fulfilled all the requirements required by Nature Portfolio journals and have been included as authors, with each significantly contributing to the design and implementation of the study. This work includes local researchers and includes findings that are locally relevant as determined by local partners. Roles and responsibilities were agreed amongst collaborators ahead of the research. This research has not been severely restricted by the setting, and has not resulted in stigmatization, incrimination, discrimination, or personal risk to participants. Local and regional research was taken into consideration in the citations.

### Reporting summary

Further information on research design is available in the Nature Portfolio Reporting Summary linked to this article.

## Data availability

Data are available for Figs. 1–7, Tables 1–3, and all Supplementary Figs. (1–11), Table (1 & 2), and Files (1 & 2). Any other data that support the conclusions of this manuscript are available from the Corresponding Author upon request. Source data are provided with this paper.

## Code availability

Python code used to process microscope images for *C. elegans* motility and viability phenotypes, as well as generate time course and dose response plots has been previously published[57].

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

## Acknowledgements

This research was funded by grants 501584 and 5003009 to Andrew Fraser from the Canadian Institute of Health Research. Xènia Serrat is supported by the European Molecular Biology Organization (EMBO ALTF 387-2021). Grants-in-Aid for Transformative Research Area (A) "Latent Chemical Space" (23H04885 to H.O.) from the Ministry of Education, Culture, Sports, Science and Technology, Japan. We thank Prof Peter Roy and Prof Erik Andersen for sharing worm strains; Prof Derek van der Kooy, Brenda Takabe and Daniel Merritt for mouse organs and use of their Dounce homogenizer; and the Peel Sausage company for the beef heart. We also thank Jason Moffat's group for letting us use their plate reader. Finally, we thank Andrew Burns and Sam Del Borrello for guidance and helpful discussion throughout.

## Author contributions

The primary drug screen was carried out by T.D., the screening of derivatives by T.D. and X.S., the in vitro mitochondrial assays were done by T.D., the cell assays were done by J.S. In the lab of I.S., the natural products were purified and provided by H.H., N.W. and H.O. and the screening in *H. polygyrus* both in vitro and in vivo by L.I. and J.K. The study was conceived of by A.F. and the paper writing and figure construction by T.D., X.S. and A.F.

## Competing interests

The authors declare no competing interests.
