## [Peer Review File · Nature Communications]

Reviewers' Comments:

Reviewer #1:

Remarks to the Author:

Davie et al: Identification of a novel family of benzimidazole species-selective Complex I inhibitors as potential anthelmintics.

The manuscript, authored by 10 scientists from 5 locations, describes the use of a rodoquinone-dependent-metabolism-*C. elegans*-assay to identify selective inhibitors withing a natural products library to identify potential nematode parasite anthelmintics.

Noteworthy: The report describes the screening of 480 distinct structural classes of compounds as well as derivatives. The authors report finding a small structurally related group of compounds, benzimidazole analogues, that they describe as having good species selectivity, inhibiting *C. elegans* Complex I; these benzimidazole analogues did not show evidence of acting on BEN-1 like albendazole. Several of these Complex I inhibitors were active against adult *H. polygyrus* in vitro but not in vivo or against L3s.

Significance: The manuscript is a significant contribution to the field: it is written clearly and contains carefully prepared figures.

The authors should address the following critiques before publication.

Major critique

The authors tested the effects of 4 of their selected benzimidazole inhibitors in vivo at 200mg/Kg but found only minor reductions in worm burden. The authors suggest that this weak effect in vivo is likely the result of poor bioavailability.

However, the authors do not test or provide information about a range of the physical properties of these compounds (logP/LogD, solubility, pKa, redox potential, molecular weight, number of atoms) that drive the pharmacokinetics of the selected inhibitors. This is relevant because cheminformatic information can be used to predict potential pharmacokinetic parameters (Lipinsky's rule of 5 or similar).

The authors do not measure any concentration of the benzimidazole inhibitors in the duodenum and small intestine and relate this to the concentrations that are effective in vitro.

The authors do not relate the presence of any stereoisomers of their compounds or their impact on observations.

Minor critiques

Reduce the number of acronyms: use rodoquinone rather than RQ; use ubiquinone rather than UQ; use benzimidazole rather than BZ; electron transport chain rather than ETC

L152: define: 'little effect'

L 190 give the selectivity ratio from IC50s of Fig 3D

L205-206 Specify 'very high' concentrations.

Check Fig 4 phenotype quantification colors and define in legend.

Fig. 7 Move to supplementary figs .

L254 Fig. not Figure.

Reviewer #2:

Remarks to the Author:

This is a comprehensive description of a fruitful and novel screen for anthelmintics, focusing on a target restricted to anaerobic GI nematodes, rodoquinone function in mitochondrial metabolism. The work builds on previous research from this group in a welcome fashion. It is well-written and thorough, and the experiments are well-described and analyzed. The lack of reported in vivo activity detracts a bit from the overall significance of this manuscript, but the science is of very high quality and the data support the importance of this pathway for anthelmintic discovery. I have one major and several minor critiques that should be resolvable by the authors.

The major concern revolves around the lack of reported in vivo activity of the most potent compounds. The authors attribute this to potentially poor bioavailability, but this is not the correct term here. The parasite lives in the GI tract, and poor bioavailability would lead to enhanced local exposure. Rather, they should postulate suboptimal exposure. It would be useful to note how rapidly the compounds work in vitro; the duration of exposure after a single dose, however high, may be too short. Secondly, the compounds may be rapidly metabolized. I recommend that the authors run a few preclinical pharmacology experiments, including microsomal stability and membrane permeability, to provide some relevant data about the lack of in vivo activity.

Minor points:

Lines 23, 46: resistance is not really a problem in human STH infections; ref. 9 addressed a tissue parasite. Resistance is a huge issue in animal health, and it is broadly accepted that anthelmintic resistance is a threat to human STH control, particularly as MDA programmes progress.

I am curious about the lack of activity of closantel et al. on *C. elegans* in aerobic conditions. Can the authors expound on why a mitotoxic ionophore should be selective in this assay? Is mitochondrial function not required in the presence of oxygen?

Finally, on line 413, the authors should know that the presence of a benzimidazole moiety does not necessarily translate to stability, ease of synthesis, etc. Assuming that further research will generate an in vivo active analogue with high potency and safety, these properties will have to be experimentally determined and may not be similar to those of other BZs.

Reviewer #3:

Remarks to the Author:

The study by Davie and colleagues addresses the urgent and highly topical subject of novel drug treatment options against soil-transmitted helminths (STH). The authors have studied a key helminth-specific pathway, the rhoquinone-dependent metabolism (RQDM) in molecular detail in various previous studies. They now build perfectly on these studies and investigate the potential of RQDM as a novel target in STH. Here they screened a library of natural products and improved derivatives for RQDM-specific mitochondrial complex I (cI) inhibition. They do so by applying a very elegant tool: KCN treatment for discrimination between oxygen-dependent cI function and RQDM-dependent cI function. The authors include tests on the target cI, on *C. elegans* worms in vitro, on *H. polygyrus* (a mouse STH) worms in vitro and in the mouse model, and they include toxicity assessments on mammalian cI and cells in vitro. They identified a highly interesting group of inhibitors that were active on in vitro cultured parasites, unfortunately not (yet) highly active on parasites grown in the mouse. Thus, these are highly relevant and noteworthy results for the field of STH, and the entire field of helminthology.

The manuscript is of high quality and carefully prepared. The language is very clear. However, there are some major points that need to be clarified regarding study design (see in particular results part below).

Introduction

Overall the introduction is very clear. A few points should receive more clarification:

- The authors should make more clear that the RQDM (malate dismutation pathway) was already described decades ago, but has only more recently been studied with novel tools and in molecular depth for specific inhibition. References are already given, but for readers outside the field of the RQDM, this is not clearly communicated.
- For readers outside the field it should also be made more clear that the RQDM is not only found in STH, but actually in all helminths, thus it could represent an overall target, certainly for STH, but also beyond. RQDM is not specific for STH, as it is currently written.
- In line with this, make clear that basically all STH (not "many", see line 24) can perform RQDM.
- Mention also that STH are part of the NTDs.

Results and Figures

The results are presented clearly and almost all data is shown. However, there are some parts to

be better clarified and there are some questions regarding study design raised below that have to be addressed to support conclusions and claims:

- Figure 1e/1f: it is not clear if this was done for this paper, or is the figure showing previously published data? If so, please refer to in figure legend.
- Lines 105 – 127 are rather introduction. Please check if parts should be moved to the introduction section.
- What was the rationale behind a screening concentration of 50uM? Is that not rather high?
- Line 178: Reference for anacardic acid missing.
- Line 182: reference missing for papaverine inhibiting cI.
- Toxicity was assessed on a human cell line, but on bovine and mouse cI. Human cI should be included as well (and bovine and mouse cell lines).
- What was the rationale behind testing inhibitors identified in the *C. elegans* model next in a STH mouse model? Why were not human STH species chosen to validate the hits from the *C. elegans* model?
- How well does the *C. elegans* model translate to the *H. polygyrus* model or STH worms? Were also some of the “non-hits” from the screening in *C. elegans* assessed in the *H. polygyrus* model? In other words, can it be ruled out that hits are missed with the *C. elegans* screen that would be active on STH?
- Why was drug activity on *H. polygyrus* not tested with the KCN-challenge model like done for *C. elegans*? Or why was the *C. elegans* screen not performed at anaerobic conditions like the *H. polygyrus* tests? Why was the method changed and not consistently kept throughout all experiments?
- How can the authors be sure that under aerobic conditions there is still RQDM and not more UQDM in adult *H. polygyrus*? Were UQ and RQ levels measured?
- Line 322: data should be shown – also if negative (as supplementary)?
- Does anaerobic culture induce slightly more RQDM in adult *H. polygyrus* as there is a little more activity of compounds? Could this be underpinned by showing expression data for *coq-2e*? Or could UQ / RQ levels be measured?
- Active compounds should not only be tested on the cI of *C. elegans*, but also the cI of *H. polygyrus* to prove the mode of action.
- What was the positive control used in the mouse model of *H. polygyrus*? This is important to show as compounds were moderately active. Without proper controls, it is difficult to draw conclusions.
- Any pharmacokinetic data available to assess why the compounds were not reaching the desired activity in the *H. polygyrus* mouse model?
- Testing on *H. polygyrus* was performed in vitro at 10 uM. How does this relate to the 200 mg/kg used in the animal model? How was the dose chosen?

Discussion

One additional point to be covered:

- Several papers in the past had mentioned that benzimidazoles like thiabendazole or mebendazole might inhibit the cI / RQDM pathway in nematodes. This is partially in line with one of the key findings in this paper and should be further discussed, including key references such as: 1–5
1. Barrowman, M. M., Marriner, S. E. & Bogan, J. A. The fumarate reductase system as a site of anthelmintic attack in *Ascaris suum*. *Biosci. Rep.* 4, 879–883 (1984).
 2. Criado Fornelio, A., Rodriguez Caabeiro, F. & Jimenez Gonzalez, A. The mode of action of some benzimidazole drugs on *Trichinella spiralis*. *Parasitology* 95, 61–70 (1987).
 3. Köhler, P. & Bachmann, R. The effects of the antiparasitic drugs levamisole, thiabendazole, praziquantel, and chloroquine on mitochondrial electron transport in muscle tissue from *Ascaris suum*. *Mol. Pharmacol.* 14, 155–163 (1978).
 4. Prichard, R. K. Mode of action of the anthelmintic thiabendazole in *Haemonchus contortus*. *Nature* 228, 684–685 (1970).
 5. Rodriguez-Caabeiro, F., Criado-Fornelio, A. & Jimenez-Gonzalez, A. A comparative study of the succinate dehydrogenase-fumarate reductase complex in the genus *Trichinella*. *Parasitology* 91, 577–583 (1985).

Methods

Overall very clear and complete, only one small point:

- Please include details on enrichment, food, and housing of mouse maintenance.

Minor points

- Make sure that drug names are consistently capitalized (or not).
- Instead of ">1000" compounds, give the precise number (abstract and introduction).
- Line 24 "RQDM" needs to be introduced.
- Omit "from here on" when giving an abbreviation in parentheses, e.g. line 36 and other.
- Missing reference line 214.
- Line 315: the mentioned reference for coq-2e splicing in *H. polygyrus* needs to be checked.
- Line 376 both sentences lack a reference.
- Line 391: name the insecticide.
- Line 664: specify: does this mean two biological replicates?
- Line 673: please state if two times independently performed.
- Color-coding of figures 4 and suppl fig 6 are unclear, please clarify in figure/legend what the colors mean and how the phenotype quantitation should be interpreted. Currently, the figure is unclear.

We thank the reviewers for their comments. In addition to addressing the great majority of text changes that they asked for (each one addressed in point by point response below), we carried out a broad comparison of many physico-chemical properties of the benzimidazole compounds with and without activity in our RQ-dependent *C. elegans* assay, yielding a new large Supplementary Figure (Supplementary Figure 8), and exposed the *H. polygyrus* adults to longer *in vitro* drug treatments (24 vs. 72-hours; Figure 7). We also thank the reviewers specifically for their comments regarding our incorrect use of 'low bioavailability' to explain the relatively weak effects *in vivo*. We note that there is also one minor change to Table 3 showing the *in vivo* testing of 4 compounds. We spotted a copy paste error in one of the control mouse worm burdens and have corrected this — the conclusions are unchanged but the efficacy values are slightly changed as a result and we apologise for this error in the previous version and are glad to have caught it. We believe the manuscript is now substantially improved as a result of addressing their suggestions and hope that they agree and thank them for their suggestions and care.

A. MAJOR CHANGES

- 1. Additional multi-panel Supplementary Figure to address the physicochemical properties of the entire set of benzimidazole compounds screened in our RQ-dependent *C. elegans* assay (Supplementary Figure 8) which shows clear distinction between hits and non-hits in multiple properties (e.g., # HBond acceptors and donors, molecular mass, # atoms and hetero atoms, logP, logSw, and TPSA).**
- 2. Two reviewers asked if longer exposure to the drugs might increase their effects and we thus tested a 72hr *in vitro* exposure for adult *H. polygyrus*. All classes of drugs had increased efficacy in this longer exposure. This is now shown in an updated Figure 7.**
- 3. We also tested the effects of several of our key benzimidazole hits against complex I from mitochondria of an expanded set of Clade V nematodes (*C. briggsae*, *P. pacificus*, and *P. hermaphrodita*) and also tested NPD8790 on mitochondria purified from HEK293 cells. We find that purified mitochondria from all the clade V nematodes show similar inhibition IC50s and that the IC50 of NPD8790 on human mitochondria extracted from HEK293 cells is similar to that for bovine and murine whole tissue-derived mitochondria.**

B. POINT BY POINT REVIEWER COMMENTS

Reviewer #1:

Major critique

The authors tested the effects of 4 of their selected benzimidazole inhibitors *in vivo* at

200mg/Kg but found only minor reductions in worm burden. The authors suggest that this weak effect in vivo is likely the result of poor bioavailability.

*Several reviewers raised this issue and we agree that 'bioavailability' is an ambiguous and unhelpful term here. We have changed the relevant text to reflect that this is more likely suboptimal exposure of the parasites to the drug and that 'bioavailability' is not the appropriate term. We also add data to show that longer drug exposure gives stronger phenotypes in the adult worms in vitro (new data added as revised Figure 7) which suggests that either increased number of doses or improved exposure might result in greater effects in vivo. The relevant text has now been changed to read "The weak in vivo effects could result from myriad causes from rapid metabolism of the compounds by the host or gut microbiota to rapid uptake and excretion by the host and thus low levels of uptake into the parasite. We also note that we delivered a single dose in vivo and perhaps multiple doses might result in a more potent response as an inhibitor of metabolism might take longer to have an effect than a paralytic like levamisole. We note that in vitro, the effects of the compounds on adult *H. polygyrus* was greater after 72hrs of treatment than after only 24hr treatment (Fig 7) which supports the idea that these may be slower acting antiparasitic drugs. We also note that mebendazole, the front line anthelmintic for human STH infections shows relatively weak efficacy in the *H. polygyrus* murine model. We suggest that medicinal chemistry studies will be needed to identify derivatives that increase the stability or delivery of these BZ-based Complex I inhibitors to the parasites in vivo."*

However, the authors do not test or provide information about a range of the physical properties of these compounds (logP/LogD, solubility, pKa, redox potential, molecular weight, number of atoms) that drive the pharmacokinetics of the selected inhibitors. This is relevant because cheminformatic information can be used to predict potential pharmacokinetic parameters (Lipinsky's rule of 5 or similar).

We agree that this should have been in the manuscript and thank the reviewer for requesting this analysis. We now extend our comparison of the benzimidazole compounds that were hits (84 compounds) vs those that showed no bioactivity (1,172 other benzimidazole compounds). We have used these analyses to generate an entire new figure (Supplementary Figure 8) and added text to summarise the findings "To gain insight into the structural requirements for benzimidazole-based compounds to act as complex I inhibitors we first compared the properties of the active and inactive benzimidazole compounds from our screen. In brief, active compounds tended to have lower molecular weights (285 Da vs. 315 Da), fewer hydrogen bond donors and acceptors (3 vs. 5.2), higher computed octanol/water coefficients (logP; 4.3 vs. 3.3), and lower topological polar surface area (TPSA; 28.1 vs. 57.7) (Supplementary Fig. 8)."

The authors do not measure any concentration of the benzimidazole inhibitors in the duodenum and small intestine and relate this to the concentrations that are effective in vitro.

That would be ideal but is a very involved experiment. We would need additional animal experiments as well as developing a mass spec assay for the compounds and their (as yet

unknown) metabolites. We hope that the reviewer agrees that this would be a large amount of work with uncertain outcomes and feel that the main findings of the study are still valid.

The authors do not relate the presence of any stereoisomers of their compounds or their impact on observations.

We used a racemic mix of stereoisomers for chiral molecules at all stages of the study since this was all that was commercially available and custom synthesis of individual stereoisomers would have been expensive. We do however add a “Stereochemistry” column in Supplementary Table 2 to highlight the 11 benzimidazole compounds in our study that have stereoisomers.

Minor critiques

Reduce the number of acronyms: use rholoquinone rather than RQ; use ubiquinone rather than UQ; use benzimidazole rather than BZ; electron transport chain rather than ETC

We had contracted these as many (e.g., ETC, UQ are fairly standard). Rather than replace all, we have just changed BZ to “benzimidazole” and RQDM to RQ-dependent metabolism throughout as these are non-standard abbreviations and we hope this is OK with the reviewer.

L152: define: ‘little effect’

We now change this to read “8 compounds that showed little effect on growth in normoxia (z-scores > -3)”

L 190 give the selectivity ratio from IC50s of Fig 3D

We now add text to give all the IC50s measured.

L205-206 Specify ‘very high’ concentrations.

Thank you this was sloppy on our part and the text now reads “until at very high concentrations (greater than 500µM)”

Check Fig 4 phenotype quantification colors and define in legend.

Thank you for spotting this, this was some pdf compilation error and we have fixed this.

Fig. 7 Move to supplementary figs .

*Since the testing of the compounds on real parasitic worms was a major additional step, and one that gave positive data, we feel strongly that we want this in the main paper and not hidden in the supplement and hope the reviewer is fine with that decision. However, we have revised Figure 7 to include *H. polygyrus* in vitro viability data for 24 and 72 hour drug treatments, and have moved 24 hour drug treatments under aerobic and anaerobic conditions to the supplementary figures (Supplementary Figure 11).*

L254 Fig. not Figure.

Thank you we have changed this.

Reviewer #2:

The major concern revolves around the lack of reported in vivo activity of the most potent compounds. The authors attribute this to potentially poor bioavailability, but this is not the correct term here. The parasite lives in the GI tract, and poor bioavailability would lead to enhanced local exposure. Rather, they should postulate suboptimal exposure.

We absolutely agree with the reviewer here and this was incorrect terminology on our part – thank you for pointing this out. We have editing the relevant sections of the text to reflect this and hope the reviewer feels that what we now say is more formally correct:

*“The weak in vivo effects could result from myriad causes from rapid metabolism of the compounds by the host or gut microbiota to rapid uptake and excretion by the host and thus low levels of uptake into the parasite. We also note that we delivered a single dose in vivo and perhaps multiple doses might result in a more potent response as an inhibitor of metabolism might take longer to have an effect than a paralytic like levamisole. We note that in vitro, the effects of the compounds on adult *H. polygyrus* was greater after 72hrs of treatment than after only 24hr treatment (Fig 7) which supports the idea that these may be slower acting antiparasitic drugs. We also note that mebendazole, the front line anthelmintic for human STH infections shows relatively weak efficacy in the *H. polygyrus* murine model. We suggest that medicinal chemistry studies will be needed to identify derivatives that increase the stability or delivery of these benzimidazole-based Complex I inhibitors to the parasites in vivo.”*

It would be useful to note how rapidly the compounds work in vitro; the duration of exposure after a single dose, however high, may be too short. Secondly, the compounds may be rapidly metabolized. I recommend that the authors run a few preclinical pharmacology experiments, including microsomal stability and membrane permeability, to provide some relevant data about the lack of in vivo activity.

We agree that extended treatment might indeed increase the strength of the effect and to that end have tested longer exposure times in vitro. We have added these data in an updated Figure 7 — they show that 72hr exposure results in more potent killing and we add text to comment on this. Regarding the pharmacology experiments, we are not set up to do these right now but agree that they must be a key part of our future analyses where we will try to identify derivatives that show improved efficacy in vivo. However for this study, we feel that identifying a new set of compounds that specifically kill nematodes when they are using RQ dependent metabolism, identifying the target and showing efficacy against parasites should be sufficient for the first paper — the manuscript is already at 7 figures and 3 tables + 11 supplemental panels and 4 tables and we hope the reviewer is willing to accept this first report as it stands in this respect.

Minor points:

Lines 23, 46: resistance is not really a problem in human STH infections; ref. 9 addressed a tissue parasite. Resistance is a huge issue in animal health, and it is broadly accepted that anthelmintic resistance is a threat to human STH control, particularly as MDA programmes progress.

We agree that full resistance has not yet been observed in any human parasite but rather that instances of reduced efficacy are increasingly reported. We have thus altered the text to reflect this and agree that we were lax in the way we wrote this before — thank you for pointing this out. The relevant sections now read: “There are few classes of anthelmintics and there is an urgent need for new anthelmintic drugs.” and “and cases of reduced efficacy have been observed in human parasites as well^{8,9}”

I am curious about the lack of activity of closantel et al. on *C. elegans* in aerobic conditions. Can the authors expound on why a mitotoxic ionophore should be selective in this assay? Is mitochondrial function not required in the presence of oxygen?

*Totally agree – it surprised us too and I am really not sure why this is. All we know is from our data: closantel and rafoxanide resulted in potent killing in our RQDM assay but NOT in aerobic conditions. We don't believe this is some oddity of *C. elegans* since both Prof Keiser and Prof Aroian's groups assessed the effects of both compounds in a large-scale drug screen in L3 stage parasites in vitro in aerobic conditions and neither saw any effects. Perhaps this is some threshold effect — in anaerobic conditions when nematodes use RQDM they may simply be far more sensitive to anything that interferes with mitochondrial function since they only use Complex I for proton pumping. We also note that FCCP, the uncoupling reagent, is far more potent in RQ-requiring anaerobic assays than aerobic conditions, which is consistent with a model where nematodes are more sensitive to compounds that perturb the mitochondrial membrane potential when they use RQ dependent metabolism.*

Finally, on line 413, the authors should know that the presence of a benzimidazole moiety does not necessarily translate to stability, ease of synthesis, etc. Assuming that further research will generate an in vivo active analogue with high potency and safety, these properties will have to be experimentally determined and may not be similar to those of other BZs.

Again we agree with the reviewer here and have altered the text to reflect this – thank you! The text now reads “given the large abundance of commercially available benzimidazole analogs and existing studies that have examined the bioavailability of benzimidazole anthelmintics, we are optimistic that improvements can be made to the in vivo activity of the NPD8790 family. We thus believe the NPD8790 family of complex I inhibitors merit further drug development and have potential to lead to new anthelmintics.”

Reviewer #3:

The manuscript is of high quality and carefully prepared. The language is very clear. However, there are some major points that need to be clarified regarding study design (see in particular results part below).

Introduction

Overall the introduction is very clear. A few points should receive more clarification:

- The authors should make more clear that the RQDM (malate dismutation pathway) was already described decades ago, but has only more recently been studied with novel tools and in molecular depth for specific inhibition. References are already given, but for readers outside the field of the RQDM, this is not clearly communicated.

*We apologise if this was not clear and have added text to reflect this — we did not mean in any way to diminish or overlook the huge progress that was made through biochemistry and beautiful mechanistic work by many groups to establish the basic pathway. We now state “This basic pathway (often also termed ‘malate dismutation’) was first described over 50 years ago in a series of elegant biochemical studies. More recently, the pathway for RQ synthesis and the key molecular switch that determines whether UQ or RQ is made were identified in the genetically tractable nematode *C. elegans* and thus we now know the way the electron transport chain is rewired, how the key electron carrier RQ is made, and many of the genes required.” and cite references appropriately.*

- For readers outside the field it should also be made more clear that the RQDM is not only found in STH, but actually in all helminths, thus it could represent an overall target, certainly for STH, but also beyond. RQDM is not specific for STH, as it is currently written.

Thank you for pointing it out. This was imprecise on our part and we have added text to clarify and correct this. The relevant section in the intro now reads “RQ is only made and used by a small number of animal species: nematodes, platyhelminths, annelids and molluscs. Since STHs make and use RQ but vertebrate hosts do not, RQ synthesis and RQDM provide a critical target that differs between host and parasite”).

- In line with this, make clear that basically all STH (not “many”, see line 24) can perform RQDM.

To be honest, there are many STH species that infect mammalian hosts and I am not sure I can make a definitive call on which of the rarer and less characterised ones use RQ. That is why I wrote “many” — I’m just not clear we have data to justify “basically all”. That may be simply our ignorance but there are so many species.

- Mention also that STH are part of the NTDs.

*We have now done so in the introduction: “Over a billion humans are infected by STHs including roundworm (*Ascaris lumbricoides*), hookworm (*Necator americanus* and *Ancylostoma duodenale*), and whipworm (*Trichuris trichiura*) and STH infections are responsible for multiple neglected tropical diseases (NTDs).”*

Results and Figures

The results are presented clearly and almost all data is shown. However, there are some parts to be better clarified and there are some questions regarding study design raised below that have to be addressed to support conclusions and claims:

- Figure 1e/1f: it is not clear if this was done for this paper, or is the figure showing previously published data? If so, please refer to in figure legend.

I am assuming that the reviewer means 1d and 1e — the data shown were newly generated for this paper but are essentially a recapitulation of data in Del Borrello et al 2019 and we have now added a note to reflect this in the text: “This was first shown in Del Borrello et al and we confirm those results here.”

- Lines 105 – 127 are rather introduction. Please check if parts should be moved to the introduction section.

We would prefer this remains here so that readers get the exact background to the drug screen just before the screen results so they can situate the data in context and hope this is ok with the reviewer.

- What was the rationale behind a screening concentration of 50uM? Is that not rather high?

This is pretty standard in initial drug screens in C. elegans where it is often challenging to get high enough drug levels into the worm to see effects — the cuticle is relatively impermeable, and C. elegans has a potent xenobiotic response to many drugs resulting in their metabolism or excretion. We have in fact tested several of our hits in a C. elegans bus-5 mutant which has increased permeability and find that their LD50 is much lower (~10X fold) but prefer not to screen directly in this strain in case of any unforeseen drug/mutant interactions. Note this is very different to drug screens in cells in culture. We prefer to screen at a higher dose and then rescreen with a full dose response curve to avoid spurious false negatives from the primary screen as much as possible. We now add a comment in the text and reference to indicate that this concentration is in line with other drug screens in C. elegans: “We conduct this primary screen at a high test compound concentration of 50µM to reduce false negatives — we note that C. elegans drug screens typically use similar primary screen concentrations as the worm has powerful xenobiotic responses that often reduce drug efficacy.”

- Line 178: Reference for anacardic acid missing.

This line referred to data generated in this paper, but a reference to Table 2 has been added to make this more clear.

- Line 182: reference missing for papaverine inhibiting cl.

This has now been added.

- Toxicity was assessed on a human cell line, but on bovine and mouse cl. Human cl should be included as well (and bovine and mouse cell lines).

Specifically for metabolic analysis, we strongly prefer using mitochondria extracted from real tissue as freshly as possible — cell lines are grown in high oxygen in artificial environments (e.g., bathed in nutrients, growth factors) which could impact their metabolic state substantially. In addition bovine heart mitochondria are standard in the mitochondrial field and are used extensively so we wanted our data to be generated in that system. However, to address some of this comment, we have now also purified mitochondria from HEK293 cells and find a very

similar IC₅₀ for NPD8790 compared to that observed in bovine and murine tissue-extracted mitochondria and now state this in the text (“In addition, we tested NPD8790 on mitochondria purified from several other nematode species (Caenorhabditis briggsae, Pristionchus pacificus and Phasmarhabditis hermaphrodita) as well as mitochondria purified from HEK293 human embryonic kidney cells and find the IC₅₀ of NPD8790 against complex I is similar for all nematode species (C. briggsae, IC₅₀ = 1.6 μM; P. pacificus, IC₅₀ = 1.2 μM; P. hermaphrodita, IC₅₀ = 3.1 μM) but human mitochondria have a greater IC₅₀ value (IC₅₀ = > 75 μM), similar to that seen in bovine and murine mitochondria (Supplementary Fig. 6).”) and hope the reviewer is satisfied with this. These data are now in an additional supplemental figure panel (Supplementary Fig. 6)

- What was the rationale behind testing inhibitors identified in the C. elegans model next in a STH mouse model? Why were not human STH species chosen to validate the hits from the C. elegans model?

Very simply – this would require human hosts! Of course every parasite is different and especially every niche and life cycle is subtly different so what may work in one mouse parasite may not work in a different mouse parasite or human parasite. But pragmatically, we cannot jump straight from worm to human so we test first in as tractable a system as we can with H. polygyrus in mice — even that is very involved. We hope the reviewer understands these practical and ethical limitations.

- How well does the C. elegans model translate to the H. polygyrus model or STH worms? Were also some of the “non-hits” from the screening in C. elegans assessed in the H. polygyrus model? In other words, can it be ruled out that hits are missed with the C. elegans screen that would be active on STH?

It is very labour intensive to screen drugs in H. polygyrus, especially in the adult stages which require mouse sacrifice to extract the worms. While for completeness sake it might be ideal to test a number of non-hits from the C. elegans screens in H. polygyrus, it is a very large additional amount of work and so given the animal use and the large workload, we cannot justify that ethically or pragmatically. We do note that the great majority of working anthelmintics have a potent effect in C. elegans so the false negative rate is not high.

- Why was drug activity on H. polygyrus not tested with the KCN-challenge model like done for C. elegans? Or why was the C. elegans screen not performed at anaerobic conditions like the H. polygyrus tests? Why was the method changed and not consistently kept throughout all experiments?

We had no access to an anaerobic hood that would allow us to do microscopy for C. elegans as is needed for the image-based RQDM assay so KCN was the sole pragmatic solution. For H. polygyrus, there was a different issue — they did not appear to tolerate KCN very well and there was considerable toxicity from KCN alone which made the assays problematic and noisy. So while it is not ideal to switch between these methods, it was done for practical reasons and we hope the reviewer understands. We have added a short text section explaining this in the Results section: “We note that we did not use anaerobic conditions in C. elegans since we could not fit

our imaging equipment into an available hood, and did not use KCN for the H. polygyrus experiments since in vitro the H. polygyrus animals were unhealthy in KCN alone.”

- How can the authors be sure that under aerobic conditions there is still RQDM and not more UQDM in adult H. polygyrus? Were UQ and RQ levels measured?

We do not formally know this — we have not directly measured UQ and RQ in H. polygyrus adults since we would have needed to sacrifice too many mice to extract the number of worms needed for mass spec analysis. However, in every helminth where RQ has been measured (a) RQ is the dominant quinone in adult stages and (b) this coincides with a switch in splicing from mostly coq-2a (UQ) to mostly coq-2e (RQ). Although we did not measure RQ and UQ in H. polygyrus adults we did look at coq-2 splicing and see a high level of coq-2e splicing in H. polygyrus adults indicating that, like all other helminths, they are also largely RQ containing and we state “Adults typically express high levels of the coq-2e isoform that is required for RQ synthesis and consistent with this, H. polygyrus adults express mostly the coq-2e isoform (67%)”. Thus while we have not done the perfect experiment, and measured RQ and UQ levels, the RNAseq data on coq-2 splicing indicate that, as expected, H. polygyrus will resemble all other helminths and be largely RQ containing in their adult stage.

- Line 322: data should be shown – also if negative (as supplementary)?

All the phenotyping of C. elegans drug effects is automated image analysis and thus we have full data for all drugs and doses. However in the case of L3 and adult parasites, the scoring is all manual and thus ‘no significant effect’ is a manual call. We thus don’t have equivalent quantitative data to show for the parasites. We have modified the text to reflect this and hope this satisfies the reviewer.

- Does anaerobic culture induce slightly more RQDM in adult H. polygyrus as there is a little more activity of compounds? Could this be underpinned by showing expression data for coq-2e? Or could UQ / RQ levels be measured?

Honestly, the marginal increase in effect of the compounds in the anaerobic conditions is very slight and we note that the worms are not particularly healthy in these conditions since they are not being fed and so there is such a large change that we are not clear that it is worth studying these unhealthy worms that lack nutrition and in vitro setting in any real depth.

- Active compounds should not only be tested on the cl of C. elegans, but also the cl of H. polygyrus to prove the mode of action.

We have now added several additional Clade V nematodes (closely related to H. polygyrus) in our analysis (Supplementary Fig. 6 and 9) and in all cases our new compounds act as complex I inhibitors. We really think it would be unlikely that they act as complex I inhibitors in a variety of Clade V nematodes and also in bovine, murine, and human cell line mitochondria but have a completely novel mechanism of action in H. polygyrus. While we agree testing against H. polygyrus would be ideal, we note that to test this would require many more mouse sacrifices to get sufficient mitochondria from the adult worms as we need a LOT of material for

mitochondrial dose response assays. We simply don't think this is worth the additional mouse sacrifice (which carry ethical considerations) and hope the reviewer agrees.

- What was the positive control used in the mouse model of *H. polygyrus*? This is important to show as compounds were moderately active. Without proper controls, it is difficult to draw conclusions.

The in vivo murine model of H. polygyrus infection is a standard assay that has been performed and published by the Keiser lab for a variety of commercially available anthelmintic drugs. For the purposes of comparison, we now reference in the text previous testing from the Keiser lab in the same H. polygyrus model for several well established commercial anthelmintics from distinct chemical classes (e.g., fenbendazole, moxidectin, and levamisole).

- Any pharmacokinetic data available to assess why the compounds were not reaching the desired activity in the *H. polygyrus* mouse model?

We indeed plan to do such experiments in our next phase of this project where we will be trying to identify more potent derivatives that also show increased in vivo effects but feel this is beyond the scope of this first study.

- Testing on *H. polygyrus* was performed in vitro at 10 uM. How does this relate to the 200 mg/kg used in the animal model? How was the dose chosen?

The rationale was that we tested a relatively low dose in vitro to identify compounds with greatest potency to prioritise them for the in vivo work. In vivo, we used a single relatively high dose in this first assay to have a good chance of efficacy for any promising compound. Relating an in vivo dose to an in vivo dose is extremely hard since it depends on pharmacokinetics and metabolism by host, parasite and microbiome and thus it is very hard to relate these in any simple fashion.

Discussion

One additional point to be covered:

- Several papers in the past had mentioned that benzimidazoles like thiabendazole or mebendazole might inhibit the cl / RQDM pathway in nematodes. This is partially in line with one of the key findings in this paper and should be further discussed, including key references such as: 1–5

1. Barrowman, M. M., Marriner, S. E. & Bogan, J. A. The fumarate reductase system as a site of anthelmintic attack in *Ascaris suum*. *Biosci. Rep.* 4, 879–883 (1984).

2. Criado Fornelio, A., Rodriguez Caabeiro, F. & Jimenez Gonzalez, A. The mode of action of some benzimidazole drugs on *Trichinella spiralis*. *Parasitology* 95, 61–70 (1987).

3. Köhler, P. & Bachmann, R. The effects of the antiparasitic drugs levamisole, thiabendazole, praziquantel, and chloroquine on mitochondrial electron transport in muscle tissue from *Ascaris suum*. *Mol. Pharmacol.* 14, 155–163 (1978).

4. Prichard, R. K. Mode of action of the anthelmintic thiabendazole in *Haemonchus contortus*. *Nature* 228, 684–685 (1970).

5. Rodriguez-Caabeiro, F., Criado-Fornelio, A. & Jimenez-Gonzalez, A. A comparative study of the succinate dehydrogenase-fumarate reductase complex in the genus *Trichinella*. *Parasitology* 91, 577–583 (1985).

We have now added several of these references but think they need to be treated with some caution. Some groups report effects on Complex I, others on Complex II, and they also report effects of levamisole, which has no effect on the electron transport chain, in their assays. We now state “We note that while some early studies suggested a possible role for benzimidazoles in affecting the electron transport chain¹⁻³, these were using ‘classic’ benzimidazoles that were subsequently all found to target microtubule polymerisation and which have no activity against complex I in our hands. Different groups reported different targets and, in addition, some of these early studies also found that levamisole (a known nicotinic acetylcholine receptor agonist that has no other known target) affected the electron transport chain¹, suggesting that these studies had a significant rate of false positives in their assays. We thus would use caution interpreting some of these early findings and suggest that our findings here are the first solid evidence of a structurally distinct group of benzimidazole-containing compounds that act as specific and potent inhibitors of complex I.”.

Methods

Overall very clear and complete, only one small point:

- Please include details on enrichment, food, and housing of mouse maintenance.

We have now given some details on housing and food and water in the Methods section.

Minor points

- Make sure that drug names are consistently capitalized (or not).

Thank you – this is now standardized throughout.

- Instead of “>1000” compounds, give the precise number (abstract and introduction).
Done.

- Line 24 “RQDM” needs to be introduced.

Thank you – this is was a bad editing mistake on our part, now fixed.

- Omit “from here on” when giving an abbreviation in parentheses, e.g. line 36 and other.

Done.

- Missing reference line 214. I’m not sure which aspect needs a reference here?

Corrected.

- Line 315: the mentioned reference for coq-2e splicing in *H. polygyrus* needs to be checked.

*It is correct – it’s the only deep transcriptome data we could find for different *H. polygyrus* stages.*

- Line 376 both sentences lack a reference.

The finding that anacardic acid inhibits Complex III and IV is this paper (data now referenced in Table 2)

- Line 391: name the insecticide.

Thank you, we do this now “We note that there is a single previous report of a compound with insecticidal activity and a benzimidazole core that is described as a putative complex I inhibitor (e.g., 1-(3,7-dimethyl-7-ethoxy-2-octenyl)-2-methylbenzimidazole)⁴⁵. However, the structure is distinct to the hits we have identified here”

- Line 664: specify: does this mean two biological replicates?

Yes and we now write “The whole experiment was performed two times i.e. two independent biological repeats.” to make this clear. Thank you.

- Line 673: please state if two times independently performed.

No, this was a single experiment. It is hard to get ethical approval for a repeat of a failed experiment — we aim to repeat at higher doses, and repeated doses in the future and are waiting for approval now.

- Color-coding of figures 4 and suppl fig 6 are unclear, please clarify in figure/legend what the colors mean and how the phenotype quantitation should be interpreted. Currently, the figure is unclear.

Thank you for spotting this, this was some pdf compilation error and we have fixed this. We have also adjusted the figure legends to describe the colour bar and clarify phenotype quantitation.

Reviewers' Comments:

Reviewer #1:

Remarks to the Author:

Revised manuscript:

Davie et al: Identification of a novel family of benzimidazole species-selective Complex I inhibitors as potential anthelmintics.

The manuscript has been revised in response to the critiques of three reviewers. All of the points raised in the critiques have been reviewed, and considered carefully by the authors. Additional appropriate information and experiments have been added to the manuscript in response. Some of the additional experiments suggested by referees required more animal experimentation which the authors argue is ethically undesirable and beyond the programmatic design of the study.

Responses to the referees' critiques were positive, appropriate and beneficial to the manuscript.

In the opinion of this referee, the manuscript is suitable for preparation for publication: The manuscript is a significant contribution to the field of study.

Authors: Review Line 366 'no other known target': levamisole is used as an inhibitor of alkaline phosphatase (Ponder and Wilkinson PMID 7024402).

Reviewer #2:

Remarks to the Author:

I appreciate the authors efforts to revise the manuscript in response to my concerns, which have been fully and satisfactorily addressed. It is my judgment that the manuscript should now be accepted for publication.

Reviewer #3:

Remarks to the Author:

All points raised by the reviewers were addressed and clarified – in the response letter to the reviewers, but also with respective adaptations in the manuscript. I highly acknowledge the additional experiments performed on human mitochondria, and the addition of a new supplementary figure 8.

The entire manuscript has been carefully re-edited and is much clearer now. Great work overall!

Some more minor details to be corrected:

- Lines 629-632: bracket not closed
- Line 942: 2 in CO₂ should be in subscript
- Make sure throughout the manuscript to have a space between number and unit (or not) – to be consistent (e.g. line 1108)
- Be also consistent with the use of the abbreviation h for hours (e.g. line 1468)